# On the role of descending drive and group III/IV muscle afferent feedback in modulating corticomotoneuronal excitability during knee-extensor exercise

Fabio G. Laginestra[1] ![ORCID], Joshua C. Weavil[2], Vincent P. Georgescu[3], Taylor S. Thurston[3], Hsuan-Yu Wan[1] ![ORCID], Nathaniel M. Birgenheier[1], Jacob E. Jessop[1], Harrison T. Finn[1] ![ORCID], Scott R. Junkins[1] and Markus Amann[1,2] ![ORCID]

[1]Department of Anesthesiology, University of Utah, Salt Lake City, UT, USA
[2]Geriatric Research, Education and Clinical Center, George E. Wahlen Department of Veterans Affairs Medical Center, Salt Lake City, UT, USA
[3]Department of Nutrition and Integrative Physiology, University of Utah, Salt Lake City, UT, USA

Handling Editors: Richard Carson & Jakob Škarabot

The peer review history is available in the Supporting Information section of this article (https://doi.org/10.1113/JP289021#support-information-section).

**Fabio G. Laginestra** obtained his PhD in Exercise Physiology at the University of Verona, Italy. He is currently a Postdoctoral Fellow in the Utah Vascular Research Laboratory at the University of Utah, under the supervision of Dr Markus Amann. His research interests revolve around physiological mechanisms underlying fatigue and the neural control of cardiorespiratory responses during physical activity.

**Abstract figure legend** Transcranial magnetic stimulation (TMS), cervicomedullary stimulation (CMS), and femoral nerve stimulation (FNS) were used to assess changes in corticomotoneuronal excitability induced by fatiguing submaximal knee-extension contractions performed in three conditions: (1) voluntarily (VOL); (2) electrically evoked (EVO); and (3) electrically evoked with pharmacological attenuation of group III/IV leg muscle afferent feedback (EVO$_{FENT}$). A control protocol (CTRL) involving only neuromuscular assessments (matched in number to those in EVO) without any fatiguing exercise was also performed. All fatiguing protocols were performed at 20% of maximal voluntary contraction and continued until quadriceps twitch torque declined by 40%. Assessments were performed during contractions at constant EMG activity before and after exercise.

**Abstract** We investigated the impact of voluntary descending drive and group III/IV leg muscle afferents on motor cortical and motoneuronal excitability during fatiguing knee-extensor exercise. Nine participants performed intermittent isometric knee extensions (20% of maximal voluntary contraction; 50 s contractions, with 10 s break): (1) voluntarily (VOL; requiring descending drive); (2) electrically evoked (EVO; without descending drive); and (3) electrically evoked with group III/IV leg muscle afferents blocked via lumbar intrathecal fentanyl (EVO$_{FENT}$). Quadriceps twitch torque ($Q_{tw}$) was quantified during the 10 s breaks. According to the study design, the pre- to post-exercise decrease in $Q_{tw}$ (~40%) was not different between trials. During constant-EMG contractions before and immediately after exercise, transcranial magnetic (TMS) and cervicomedullary stimulations were administered to elicit conditioned (preceded by a conditioning TMS pulse) vastus lateralis motor-evoked (cMEP) and cervicomedullary motor-evoked (cCMEP) potentials. Following VOL, cMEP and cCMEP were significantly decreased by $70 \pm 16$ and $82 \pm 19\%$ (both $P < 0.001$), and the cMEP/cCMEP ratio was significantly increased. Without affecting cMEP/cCMEP, EVO significantly decreased cMEP (by $40 \pm 23\%$) and cCMEP (by $26 \pm 24\%$, both $P \leq 0.012$), but these changes were significantly smaller compared with VOL ($P < 0.001$). The corticomotoneuronal consequences of EVO and EVO$_{FENT}$ were not different ($P > 0.964$). These findings suggest that voluntary fatiguing knee-extension exercise enhances motor cortical excitability but compromises motoneuronal excitability, with the combined effect of an overall depression of the corticomotoneuronal pathway. Mechanisms associated with descending drive appear to mediate the increase in motor cortical excitability and largely, but not exclusively, account for the inhibition of the motoneuron pool during fatiguing knee-extensor exercise. Group III/IV muscle afferents do not contribute to the cortical and motoneuronal excitability changes during this exercise modality.

(Received 5 April 2025; accepted after revision 22 November 2025; first published online 17 December 2025)

**Corresponding author** F. G. Laginestra: Department of Anesthesiology, University of Utah, Salt Lake City, UT 84148, USA. Email: fabio.laginestra@utah.edu

## Key points

- We investigated the impact of voluntary descending drive and group III/IV leg muscle afferent feedback on corticomotoneuronal excitability during fatiguing knee-extensor exercise (KE).
- Transcranial and cervicomedullary stimulations were used in conjunction with submaximal voluntary and electrically evoked KE (matched for end-exercise quadriceps fatigue) and pharmacological blockade of group III/IV leg muscle afferent feedback.
- Voluntary KE enhanced motor cortical but compromised motoneuronal excitability, with the net effect of an overall depression of the corticomotoneuronal pathway. Electrically evoked KE impaired motoneuronal excitability but not motor cortical excitability, resulting in an overall depression of the corticomotoneuronal pathway. However, this impact was smaller compared with voluntary KE.
- Motor cortical and motoneuronal changes were similar during electrically evoked KE performed with intact and pharmacologically blocked group III/IV muscle afferents.

- Mechanisms associated with voluntary descending drive facilitate the motor cortex and largely, but not exclusively, account for the decrease in motoneuronal excitability during fatiguing KE. Group III/IV muscle afferents do not contribute to these changes during this exercise modality.

## Introduction

The excitability of the corticomotoneuronal pathway, including the motor cortex and spinal motoneurons, changes with fatiguing exercise in humans. Although the response of motoneurons to a given synaptic input decreases during both locomotor (Weavil et al., 2016) and single-joint (Finn et al., 2018; McNeil et al., 2009) exercise, the cortical response is different between the two exercise modalities, with locomotion decreasing (Sidhu et al., 2018) and single-joint contractions facilitating (McNeil et al., 2009; Søgaard et al., 2006; Taylor et al., 2000) excitability. Although the exact mechanisms determining these changes remain unknown, descending neural drive and group III/IV muscle afferent feedback have been considered to play a role.

Despite little available data on their role in cortical excitability (Khaslavskaia & Sinkjaer, 2005; Luft et al., 2005), factors associated with descending drive can compromise the excitability of motoneurons during fatiguing muscle contractions. Strong support for this postulate comes from a study on anaesthetized and hindlimb-denervated cats (L6–S1 dorsal root cut; no influence of sensory muscle afferents on motoneurons), showing that electrical stimulation of the motoneuron soma causes hyperpolarization, resulting in progressive decreases in the excitability of motoneurons innervating hindlimb extensor muscles (Kernell & Monster, 1982). Human investigations addressing the role of descending drive have traditionally compared motoneuronal changes after electrically evoked (i.e., no descending drive) and voluntary (requiring descending drive) upper-limb muscle contractions. The outcomes are, however, conflicting with some studies showing that descending drive was necessary to observe fatigue-induced decrease in motoneuron excitability (D'Amico et al., 2020; Gandevia et al., 1999; Khan et al., 2016), whereas others have shown decreased excitability also with electrically evoked contractions (Butler & Thomas, 2003), suggesting that motoneuron activation by antidromic or reflex excitatory input and feedback from group III/IV afferents can play a role. This is supported by recent work showing that both electrically evoked and voluntary high-intensity fatiguing dorsiflexion contractions reduce maximal motor-unit firing rates (Zero et al., 2025). Although this leaves the role of descending drive in mediating fatigue-induced changes in upper-limb muscles unclear, its influence on corticomotoneuronal excitability of locomotor muscles is unknown.

Feedback from mechano- and metabosensitive group III/IV muscle afferents has also been considered a determinant of fatigue-induced changes in motoneuron output (Bigland-Ritchie et al., 1986; Garland & Kaufman, 1995). However, although animal investigations revealed facilitatory effects on the motor cortex (Luft et al., 2005) and an inhibitory effect on motoneurons controlling hindlimb muscles (Kniffki et al., 1981; Schomburg et al., 1999), their influence remains unclear in the exercising human. The few studies evaluating their effect during human locomotion suggest that group III/IV muscle afferents inhibit motor cortical excitability without affecting motoneurons (Sidhu et al., 2017, 2018). During single-joint exercise, studies using postexercise circulatory occlusion report no effect of metabo-nociceptive group III/IV muscle afferents on motoneuronal or cortical excitability of upper (Butler et al., 2003; Taylor et al., 1996) and lower (Kennedy et al., 2016) limb muscles. However, other single-joint studies using postexercise circulatory occlusion or intramuscular hypertonic saline to raise metabo-nociceptive group III/IV-mediated feedback suggest that these sensory neurons depress (Martin et al., 2006) or facilitate (Martin et al., 2008) elbow extensor motoneurons, facilitate elbow flexor motoneurons (Martin et al., 2006, 2008) and inhibit cortical excitability (Martin et al., 2008). Given the limb-specific differences in the strength of cortico-motoneuronal projections (Brouwer & Ashby, 1990) and the fact that postexercise circulatory occlusion and hyper-tonic saline isolate a particular subset of sensory neurons that is likely not to be active during freely perfused exercise (Kniffki et al., 1981; Light et al., 2008; Pollak et al., 2014), the influence of group III/IV muscle afferents on cortical and spinal excitability during single-joint leg exercise remains unclear in humans.

It was therefore the primary purpose of this investigation to evaluate the effect of descending drive on corticomotoneuronal excitability of knee extensors by comparing the consequences of voluntary *vs.* electrically evoked quadriceps exercise. Given that motor activity ongoing during the excitability assessment can facilitate the measurements and thus mask the true impact, we used an experimental technique that reduces this confounding influence (McNeil et al., 2009). Specifically, we quantified both motor-evoked potentials (MEPs) and cervicomedullary motor-evoked potentials (CMEPs) during the short interruption of motor activity (i.e., the silent period) that follows a single transcranial magnetic

stimulus to the motor cortex (Finn et al., 2018; McNeil et al., 2009; Sidhu et al., 2018). Because we found that descending drive only accounted for ∼70% of the decrease in motoneuron excitability, we tested the hypothesis that group III/IV muscle afferents contribute to the decrease in motoneuron excitability during electrically evoked exercise.

## Methods

### Ethical approval

Nine (five female and four male) healthy, recreationally active volunteers (age, $25 \pm 5$ years; body mass, $65 \pm 8$ kg; height, $170 \pm 11$ cm) participated in this study. Written informed consent was obtained from each participant. All experimental procedures were approved by the University of Utah and Salt Lake City Veterans Affairs Medical Center Institutional Review Boards (IRB approval number: 60613) and conformed to the *Declaration of Helsinki* (without registration of the study in a research database).

### Torque and EMG recordings

Knee-extension force was measured using a calibrated linear strain gauge (MLP 300; Transducer Techniques, Temecula, CA, USA) attached to a Velcro cuff fixed ∼2 cm above the lateral malleolus of the right leg. Quadriceps torque was calculated by multiplying the knee-extension force by the lever arm (distance from the centre of the patella to the lateral malleolus) and the sine of the knee angle. EMG was recorded with Ag/AgCl surface electrodes (Danlee Medical Products, Syracuse, NY, USA) placed over the muscle belly of the vastus lateralis (VL; 20 mm interelectrode distance). Prior to electrode placement, the skin was shaved, lightly abraded with sandpaper and cleaned with an alcohol swab. EMG electrode placement was recorded on the first visit and marked for replication in subsequent sessions. EMG signals were amplified (gain, $1000\times$; 1901, Cambridge Electronic Design, Cambridge, UK), band-pass filtered (10–1000 Hz; 1901, Cambridge Electronic Design) and converted from analog to digital using a 16-bit Micro 1401 mk-II and Spike 2 data collection software (Cambridge Electronic Design) via custom-written scripts. The sampling rate was set at 2000 Hz.

### Stimulations and neuromuscular function

Three forms of stimulation were applied in this study: (1) electrical femoral nerve stimulation (FNS); (2) transcranial magnetic stimulation (TMS); and (3) electrical cervicomedullary stimulation (CMS).

**Femoral nerve stimulation.** The femoral nerve was stimulated with the anode fixed on the greater trochanter and the cathode positioned on the femoral nerve (located in the femoral triangle). To determine the position at which FNS elicited the highest compound muscle action potential (M-wave) in the VL and quadriceps twitch torque ($Q_{tw}$), low-intensity single-pulse stimuli (200 μs pulse width; 75–100 mA) were delivered using a cathode probe connected to a constant-current stimulator (model DS7AH, Digitimer Ltd, Welwyn Garden City, UK). Once the optimal position was established, the cathode was fixed in this location. Thereafter, at rest, the stimulation intensity was increased in 25 mA increments until the size of the M-wave remained unchanged despite increases in intensity (i.e., maximal M-wave; $M_{max}$). The final stimulation intensity was then set at 130% of the intensity eliciting $M_{max}$ ($181 \pm 10$ mA).

**Transcranial and cervicomedullary stimulations.** A double cone coil (diameter 130 mm) attached via a BiStim unit to two Magstim 200 stimulators (The Magstim Company Ltd, Dyfed, UK) was used to elicit motor-evoked potentials (MEPs) in the VL. The optimal coil position (posterior to anterior direction of current flow in the motor cortex) was determined in order to activate preferentially the portion of the left motor cortex with the largest representation of the quadriceps muscle. This location was marked directly on the scalp for accurate placement throughout the session. At the beginning of every session, participants performed three quadriceps maximal voluntary contractions (MVCs), from which rectified EMG was calculated and averaged. For the determination of stimulation intensities and during the successive TMS and CMS procedures, all participants were instructed to perform quadriceps contractions corresponding to 20% of the maximum EMG activity recorded during MVCs. Real-time visual feedback based on the rectified and smoothed (500 ms window) EMG was provided to guide the subject in maintaining the target level. One Magstim stimulator was used to deliver the conditioning stimulus, and the other stimulator was used to deliver the test stimulus evoking a conditioned MEP (cMEP). The intensity of the conditioning stimulus (50–90% stimulator output) was set to produce a silent period of ∼200 ms in the EMG burst. To ensure that the test stimulus was delivered during a period of complete EMG quiescence, the interstimulus interval was determined individually for each participant. Specifically, if residual EMG activity was observed at 100 ms, the interval was adjusted (always >80 ms) until no activity was visible, then maintained consistently across all trials for that participant (mean $\pm$ SD: $93 \pm 6$ ms). Based on previous literature, interstimulus intervals of >80 ms should produce comparable outcomes whilst

reducing the likelihood of spinal inhibition (Yacyshyn et al., 2016). In three of the six paired-pulse stimulations, the test TMS was replaced by CMS, evoking a conditioned cervicomedullary motor-evoked potential (cCMEP). An electrical percutaneous stimulator (D-185 mark IIa, Digitimer Ltd) was used to activate the cervicomedullary junction at the back of the neck to elicit cCMEPs in the VL. This was achieved by passing a high-voltage pulse (100 µs pulse width) between a set of self-adhesive electrodes attached to the skin in the groove between the mastoid processes and the occiput (with cathode electrode on the left, contralateral to the exercising knee extensors). The stimulation intensity ($578 \pm 17$ V) of the test stimulus was set to evoke a cCMEP of 20% $M_{max}$. The test stimulus intensity for TMS was then set to elicit a cMEP response that was approximately equivalent in size to the cCMEP response (test TMS intensity = $65 \pm 12\%$ stimulator output). TMS and CMS were also delivered in isolation (i.e., without a conditioning stimulus) and at the same intensity used for test stimuli, with the purpose of evoking unconditioned MEPs (uMEP) and unconditioned CMEPs (uCMEP).

**Assessment of neuromuscular quadriceps function.** Participants were seated on a custom-built chair with full back support, such that the hip and knee angles were $\sim 120°$ and $\sim 90°$, respectively. They then performed a 3 s quadriceps MVC, during which an FNS was delivered to evoke a superimposed twitch (SIT) followed by another FNS to evoke a potentiated quadriceps twitch at rest ($Q_{tw}$). Voluntary activation (VA; as a percentage) was calculated as follows (Merton, 1954): $VA = (1 - SIT/Q_{tw}) \times 100$.

### Experimental protocol

Each participant attended five sessions on different days, with each visit separated by a minimum of 72 h. During the first visit, the participants were familiarized with various procedures included in the study. At the beginning of all the experimental sessions, neuromuscular assessment of quadriceps function was conducted as described above (see '*Assessment of neuromuscular quadriceps function*'). Immediately after, excitability measures (i.e., cCMEP, uCMEP, cMEP, uMEP and $M_{max}$) were performed during a $\sim 15$–20 s contraction corresponding to 20% of the maximal EMG quantified during previous MVCs. The order of the stimulations was randomized during each set and delivered every $\sim 3$–5 s (Sidhu et al., 2012). Separated by 4 min, three sets of corticomotoneuronal assessments were performed, and results were averaged to establish baseline values.

During the next two experimental visits, two different fatiguing protocols (separated by a minimum of 72 h) were performed. Specifically, quadriceps contractions corresponding to 20% MVC were either performed voluntarily (i.e., requiring descending drive, VOL) or electrically evoked (i.e., no descending drive, EVO). Each protocol entailed 1 min contraction cycles consisting of a 50 s isometric contraction and a 10 s break, during which MVC torque and $Q_{tw}$ were quantified. Contraction cycles were repeated until $Q_{tw}$ decreased by $\sim 40\%$ from baseline. The order of conditions was not randomized to ensure that participants could achieve the target level of fatigue in the VOL condition first, which is typically the most demanding, because some participants might be unable to reach a 40% reduction in $Q_{tw}$. During EVO, a peripheral nerve stimulator (model DS7AH, Digitimer Ltd) was used to stimulate the femoral nerve (Hureau et al., 2018) (frequency, 40 Hz; pulse width, 200 µs) and elicit the 50 s tetanic quadriceps contraction. The stimulation intensity during EVO was adjusted manually to ensure that the target torque was maintained throughout the protocol (start of protocol, $27 \pm 6$ mA; end of protocol, $49 \pm 15$ mA). A graphical representation and other practical aspects of the two different fatiguing modalities can be found elsewhere (Laginestra et al., 2022). During the next experimental visit, the EVO protocol was repeated after group III/IV leg muscle afferent feedback was pharmacologically attenuated via the intrathecal injection of fentanyl ($EVO_{FENT}$). Finally, in a separate session designed to isolate and evaluate the corticomotoneuronal impact of the short MVCs involved in the neuromuscular function assessments conducted every minute throughout the protocol, each participant repeated EVO, but without the electrically evoked contractions (CTRL); pre/post changes in cortical and motoneuronal excitability were assessed as described above.

During each of the experimental visits, the post-exercise assessment of corticomotoneuronal excitability was initiated within 3–5 s after the fatiguing bout was terminated.

### Intrathecal fentanyl injection

After the assessment of baseline corticomotoneuronal and neuromuscular function during $EVO_{FENT}$, 0.5 mL of fentanyl (concentration, 50 µg/mL) was delivered intrathecally at the vertebral interspace L3–L4 (Amann et al., 2009). To ensure that the effects of fentanyl were limited to attenuating muscle afferent feedback from the lower limbs, we quantified the minute ventilation $\dot{V}_E$ in litres per minute) during constant-load arm cranking (15 and 30 W, 3 min each; Monark–Crescent AB, Varberg, Sweden) performed before and after fentanyl administration, as previously described (Iannetta et al., 2024; Thurston et al., 2023). Given that binding of fentanyl to medullary opioid receptors decreases the ventilatory response to upper-limb exercise (Amann et al., 2010), a similar $\dot{V}_E$

during arm cranking before and after the injection would exclude the possibility of a direct effect of fentanyl on brain opioid receptors. Therefore, ventilatory responses to arm cycling exercise performed before and after fentanyl administration were assessed on an individual basis. Reductions of >2SD from the breath-by-breath recording of $\dot{V}_E$ during the final minute of arm cycling in CTRL conditions were considered evidence of a cephalad migration of fentanyl.

### Data analysis

All data were analysed offline using Spike 2 software (Cambridge Electronic Design). Areas of MEPs, CMEPs (both conditioned and unconditioned) and $M_{max}$ were measured between cursors placed to encompass all phases of the evoked potentials. To account for potential activity-dependent changes in muscle sarcolemma excitability, MEPs and CMEPs were normalized for $M_{max}$. To quantify excitability changes at the cortical level, MEPs were normalized for CMEPs (i.e., MEP/CMEP; as a percentage). The duration of the silent period was considered as the time interval from the TMS pulse eliciting uMEP to the return of the voluntary EMG (Sidhu et al., 2018). Baseline indexes of neuromuscular function (MVC, VA and $Q_{tw}$) and motor pathway excitability/inhibition (uMEP, uCMEP, cMEP, cCMEP, cMEP/cCMEP and silent period) were averaged over three sets of assessments performed pre-exercise. Postexercise values were captured immediately after the termination of exercise and expressed as a percentage change from pre-exercise values. To ensure that EMG activity was comparable between baseline and postexercise testing contractions, the root mean square over the 200 ms EMG before every stimulation was analysed and averaged. Furthermore, the mean of the 500 ms of torque preceding each stimulation was also averaged to assess potential differences in the exercise-induced changes in the torque associated with 20% EMG contractions. To determine the exercise-induced increase in EMG during VOL, EMG activity was quantified as the root mean square of the first 20 s of the first fatiguing contraction and for the last 20 s of each successive contraction until task cessation.

### Statistical analysis

Statistical analyses were performed in R (v.4.5.1; R Core Team) using the packages lme4, lmerTest, emmeans and clubSandwich. Data from each experimental condition were analysed using linear mixed-effects models with random intercepts for participants to account for within-participant dependencies. For each outcome variable, we computed the change from baseline expressed as the percentage change, apart from voluntary activation (as a percentage) and silent period (in milliseconds). Baseline cMEP and cCMEP size expressed as a percentage of $M_{max}$ were also compared. The fixed factor in all models was condition (four levels: VOL, EVO, EVO_FENT and CTRL). Only the average torque during the fatiguing contractions and the number of contractions needed to reach task failure were run on three levels (VOL, EVO and EVO_FENT). Estimated marginal means for each condition were computed, and robust standard errors were used to address potential heteroscedasticity. Two sets of statistical tests were performed: (1) tests of deviation from zero, to determine whether the mean change within each condition differed significantly from baseline; and (2) pairwise comparisons between conditions, to assess differences in the magnitude of change across the four experimental conditions. Pairwise contrasts were conducted (VOL *vs.* EVO, EVO *vs.* EVO_FENT and EVO *vs.* CTRL) with *P*-values successively corrected using the Sidak procedure. Ventilatory data before and after fentanyl injection were compared using Student's paired *t*-tests corrected for two comparisons, also using the Sidak procedure. Estimated marginal means (*B*) and 95% confidence intervals (CIs) are reported throughout. Although the model provides the SEM for statistical purposes, we reported data as the mean ± SD for clarity. Graphical presentation of the results was performed using Graphpad Prism v.8.0. Statistical significance was set at $P < 0.050$.

## Results

### Arm cranking exercise

The $\dot{V}_E$ response to arm cycling after fentanyl injection was within 2SD of that recorded prior to drug injection in all participants. Therefore, a direct effect of fentanyl on cerebral opioid receptors was excluded. The group ventilatory response to arm cranking performed before and after the administration of fentanyl was not different at 15 W (pre, 24 ± 2 L/min; post, 24 ± 6 L/min; $P = 0.965$) or 30 W (pre, 31 ± 3 L/min; post, 30 ± 4 L/min; $P = 0.691$).

### Exercise-induced changes in neuromuscular function and $M_{max}$

The average torque produced during the fatiguing contractions was not different between conditions (VOL, 20.3 ± 3.4% MVC; EVO, 20.5 ± 3.2% MVC; EVO_FENT, 20.1 ± 4.2% MVC; $P = 0.887$). The $Q_{tw}$ decreased significantly from pre- to postexercise during all trials (VOL: from 45 ± 9 to 25 ± 4 N m, $B = -44\%$, $P < 0.001$, 95% CI [−47, −42]; EVO: from 44 ± 12 to 24 ± 6 N m, $B = -45\%$,

$P < 0.001$, 95% CI [−48, −42]; EVO$_{FENT}$: from 48 ± 9 to 26 ± 3 N m, $B = −47\%$, $P < 0.001$, 95% CI [−50, −43]), but not in CTRL (from 45 ± 10 to 45 ± 11 N m, $B = 0\%$, $P = 0.999$, 95% CI [−4, 5]). Although the reduction in $Q_{tw}$ did, by design, not differ between VOL and EVO or between EVO and EVO$_{FENT}$, it was different between EVO and CTRL (Fig. 1*A* and *D*).

The target decrease in $Q_{tw}$ was reached after 7 ± 1 voluntary contractions during VOL and after 6 ± 3 and 5 ± 3 electrically evoked contractions during EVO and EVO$_{FENT}$, respectively ($P = 0.202$).

The MVC fell significantly from pre- to postexercise during all trials (VOL: from 188 ± 63 to 104 ± 44 N m, $B = −44\%$, $P < 0.001$, 95% CI [−53, −36]; EVO: from 192 ± 69 to 118 ± 46 N m, $B = −40\%$, $P < 0.001$, 95% CI [−46, −32]; EVO$_{FENT}$: from 195 ± 74 to 120 ± 47 N m, $B = −39\%$, $P < 0.001$, 95% CI [−44, −33]), but not in CTRL (from 192 ± 69 to 188 ± 64 N m, $B = −1\%$, $P = 0.958$, 95% CI [−5, 3]). The reduction in MVC was

not different between the three fatiguing protocols (Fig. 1*B* and *E*). The VA decreased significantly from pre- to post-exercise during VOL (from 94 ± 4 to 81 ± 7%, $B = −13\%$, $P < 0.001$, 95% CI [−18, −8]), but remained unaltered during EVO (from 93 ± 4 to 90 ± 6%, $B = −3\%$, $P = 0.247$, 95% CI [−7, 0]), EVO$_{FENT}$ (from 94 ± 3 to 91 ± 6%, $B = −3\%$, $P = 0.253$, 95% CI [−6, 0]) and CTRL (from 93 ± 5 to 90 ± 4%, $B = −3\%$, $P = 0.362$, 95% CI [−7, −1]) (Fig. 1*C* and *F*). The $M_{max}$ area (VOL: from 0.08 ± 0.03 to 0.09 ± 0.03 mV s, $B = 4\%$, $P = 0.404$, 95% CI [−1, 9]; EVO: from 0.08 ± 0.03 to 0.09 ± 0.03 mV s, $B = 6\%$, $P = 0.538$, 95% CI [−3, 15]; EVO$_{FENT}$: from 0.09 ± 0.04 to 0.09 ± 0.03 mV s, $B = −1\%$, $P = 0.912$, 95% CI [−5, 3]; CTRL: from 0.07 ± 0.02 to 0.07 ± 0.02 mV s, $B = 0\%$, $P > 0.999$, 95% CI [−3, 2]) or amplitude (VOL: from 11.4 ± 3.4 to 11.4 ± 2.9 mV, $B = 1\%$, $P = 0.996$, 95% CI [−3, 5]; EVO: from 11.7 ± 4.0 to 11.7 ± 3.9 mV, $B = 1\%$, $P = 0.997$, 95% CI [−6, 8]; EVO$_{FENT}$: from 12.3 ± 4.2 to 12.3 ± 4.2 mV, $B = 1\%$, $P = 0.992$, 95% CI [−6, 9]; CTRL: from 12.7 ± 4.5

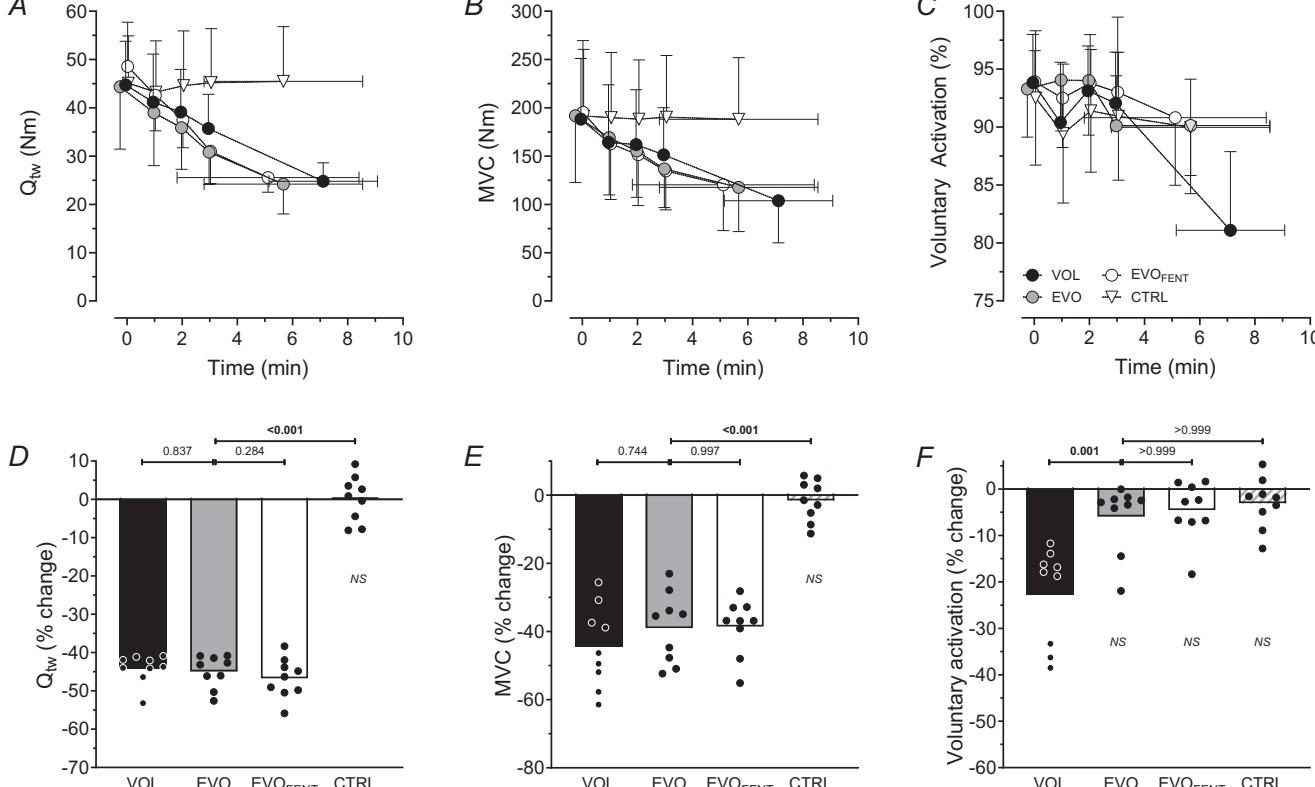

**Figure 1. Exercise-induced changes in neuromuscular function**
Exercise-induced changes in potentiated quadriceps twitch torque ($Q_{tw}$; *A* and *D*), maximal voluntary contraction (MVC) torque (*B* and *E*) and voluntary activation (*C* and *F*). Top row, progressive changes throughout each trial. Bottom row, pre- to postexercise changes. In the top row, data are shown for the first 3 min, during which all participants had data available (one point per minute). The final point represents the mean value at task termination time for each individual participant (±SD across participants). In *A–C*, the first three time points are staggered to improve clarity of the graphical illustration. Other abbreviations: CTRL, control protocol; EVO, electrically evoked contractions; EVO$_{FENT}$, electrically evoked contractions + fentanyl; NS, not significantly different from baseline; and VOL, voluntary contractions.

to $12.7 \pm 4.5$ mV, $B = -1\%$, $P = 0.950$, 95% CI $[-3, 2]$) did not change in any trial, and there were no differences between trials ($M_{max}$ area, all $P \geq 0.142$; $M_{max}$ amplitude, all $P \geq 0.925$).

The background EMG activity before stimulation did not change from pre- to postexercise (VOL: from $0.18 \pm 0.06$ to $0.19 \pm 0.07$ mV, $B = 6\%$, $P = 0.673$, 95% CI $[-4, 15]$; EVO: from $0.18 \pm 0.06$ to $0.19 \pm 0.07$ mV, $B = 4\%$, $P = 0.558$, 95% CI $[-2, 11]$; EVO$_{FENT}$: from $0.18 \pm 0.08$ to $0.17 \pm 0.08$ mV, $B = -2\%$, $P = 0.982$, 95% CI $[-10, 6]$; CTRL: from $0.18 \pm 0.07$ to $0.17 \pm 0.07$ mV, $B = -1\%$, $P = 0.932$, 95% CI $[-4, 2]$). Also, EMG activity expressed as a percentage of pre was not different between conditions (VOL, $106 \pm 14\%$; EVO, $104 \pm 10\%$; EVO$_{FENT}$, $98 \pm 12\%$; CTRL, $99 \pm 4\%$; all $P \geq 0.356$). The torque associated with the 20% EMG contractions decreased similarly from pre- to postexercise in all fatiguing protocols (VOL: from $59 \pm 17$ to $42 \pm 10$ N m, $B = -28\%$, $P < 0.001$, 95% CI $[-35, -21]$; EVO: from $61 \pm 16$ to $48 \pm 14$ N m, $B = -21\%$, $P < 0.001$, 95% CI $[-29, -14]$; EVO$_{FENT}$: from $61 \pm 15$ to $49 \pm 16$ N m, $B = -22\%$, $P < 0.001$, 95% CI $[-28, -16]$), but not in CTRL (from $58 \pm 13$ to $57 \pm 13$ N m, $B = -2\%$, $P = 0.810$, 95% CI $[-6, 2]$). EMG activity during VOL increased from $18 \pm 6$ to $40 \pm 9\%$ of MVC$_{EMG}$ ($P < 0.001$) from the first contraction to exercise cessation.

### Unconditioned responses (normalized for *M*<sub>max</sub>)

Unconditioned MEPs did not change from baseline to end-exercise in any trial (VOL: from $53 \pm 28$ to $50 \pm 21\%$, $B = 2\%$, $P > 0.999$, 95% CI $[-21, 25]$; EVO: from $53 \pm 27$ to $55 \pm 25\%$, $B = 9\%$, $P = 0.845$, 95% CI $[-13, 30]$; EVO$_{FENT}$: from $49 \pm 27$ to $47 \pm 25\%$, $B = -3\%$, $P = 0.995$, 95% CI $[-20, 14]$; CTRL: from $56 \pm 34$ to $50 \pm 26\%$, $B = -7\%$, $P = 0.804$, 95% CI $[-22, 8]$). No differences were observed in uMEP between conditions (all $P \geq 0.213$). Likewise, uCMEP did not change from baseline to end-exercise (VOL: from $31 \pm 13$ to $32 \pm 13\%$, $B = 6\%$, $P = 0.870$, 95% CI $[-8, 20]$; EVO: from $30 \pm 14$ to $30 \pm 15\%$, $B = -1\%$, $P > 0.999$, 95% CI $[-16, 13]$; EVO$_{FENT}$: from $27 \pm 11$ to $25 \pm 12\%$, $B = -6\%$, $P = 0.858$, 95% CI $[-20, 8]$; CTRL: from $24 \pm 12$ to $25 \pm 11\%$, $B = 4\%$, $P = 0.988$, 95% CI $[-16, 24]$). No differences were observed in uCMEP between conditions (all $P \geq 0.875$). Finally, uMEP/uCMEP did not change from baseline to end-exercise in any trial (VOL: from $192 \pm 121$ to $168 \pm 85\%$, $B = -1\%$, $P > 0.999$, 95% CI $[-23, 22]$; EVO: from $196 \pm 118$ to $211 \pm 102\%$, $B = 16\%$, $P = 0.643$, 95% CI $[-10, 42]$; EVO$_{FENT}$: from $189 \pm 99$ to $196 \pm 107\%$, $B = 6\%$, $P = 0.973$, 95% CI $[-16, 27]$; CTRL: from $251 \pm 148$ to $225 \pm 122\%$, $B = -4\%$, $P = 0.996$, 95% CI $[-29, 21]$). No differences were observed in uMEP/uCMEP between conditions (all $P \geq 0.210$).

### Conditioned responses (normalized for *M*<sub>max</sub>) and silent period

Representative traces of the EMG responses to paired stimuli are illustrated in Fig. 2. At baseline, cMEPs were not different between experimental conditions (all $P \geq 0.088$). Conditioned MEPs decreased significantly from baseline to end-exercise in all trials (VOL: from $23.2 \pm 19.6$ to $8.7 \pm 9.7\%$, $B = -70\%$, $P < 0.001$, 95% CI $[-81, -58]$; EVO: from $18.9 \pm 14.2$ to $11.1 \pm 8.4\%$, $B = -40\%$, $P < 0.001$, 95% CI $[-56, -24]$; EVO$_{FENT}$: from $12.2 \pm 7.6$ to $8.4 \pm 6.8\%$, $B = -37\%$, $P = 0.001$, 95% CI $[-54, -20]$) but CTRL (CTRL: from $14.2 \pm 8.8$ to $12.4 \pm 7.1\%$, $B = -10\%$, $P = 0.391$, 95% CI $[-22, 3]$) (Fig. 3*A*). Conditioned MEPs decreased significantly more in VOL than in EVO, and no differences were found between EVO and EVO$_{FENT}$ (Fig. 3*A*). At baseline, cCMEPs were not different between experimental conditions (all $P \geq 0.514$). Conditioned CMEPs decreased significantly from baseline to end-exercise during VOL, EVO and EVO$_{FENT}$ (VOL: from $16.0 \pm 9.2$ to $2.2 \pm 1.9\%$, $B = -82\%$, $P < 0.001$, 95% CI $[-95, -68]$; EVO: from $14.8 \pm 8.2$ to $11.6 \pm 8.3\%$, $B = -26\%$, $P = 0.012$, 95% CI $[-42, -9]$; EVO$_{FENT}$: from $14.1 \pm 7.8$ to $10.5 \pm 8.0\%$, $B = -31\%$, $P = 0.004$, 95% CI $[-48, -14]$), but not during CTRL (from $12.6 \pm 7.3$ to $13.8 \pm 8.6\%$, $B = 6\%$, $P = 0.844$, 95% CI $[-7, 19]$) (Figs 3*B* and 4). The decline in VOL was significantly more pronounced than in EVO, and there were no differences between EVO and EVO$_{FENT}$ (Fig. 3*B*). The cMEP/cCMEP ratio increased significantly during VOL (from $139 \pm 57$ to $349 \pm 255\%$, $B = 142\%$, $P = 0.021$, 95% CI $[46, 238]$) but remained unchanged during EVO (from $128 \pm 45$ to $119 \pm 78\%$, $B = -9\%$, $P = 0.978$, 95% CI $[-45, 27]$), EVO$_{FENT}$ (from $88 \pm 25$ to $88 \pm 50\%$, $B = -2\%$, $P > 0.999$, 95% CI $[-30, 25]$) and CTRL (from $125 \pm 58$ to $115 \pm 69\%$, $B = -9\%$, $P = 0.943$, 95% CI $[-36, 18]$) (Fig. 3*C*). The increase in VOL was significantly higher than in EVO, and no differences were found between EVO and EVO$_{FENT}$ (Fig. 3*C*). Finally, the silent period increased significantly during VOL (from $283 \pm 36$ to $321 \pm 47$ ms, $B = 38$ ms, $P < 0.001$, 95% CI $[24, 51]$) but remained unchanged during EVO (from $281 \pm 43$ to $294 \pm 45$ ms, $B = 13$ ms, $P = 0.203$, 95% CI $[0, 26]$), EVO$_{FENT}$ (from $287 \pm 47$ to $291 \pm 49$ ms, $B = 4$ ms, $P = 0.749$, 95% CI $[-4, 12]$) and CTRL (from $234 \pm 57$ to $247 \pm 71$ ms, $B = 13$ ms, $P = 0.783$, 95% CI $[-13, 39]$). The lengthening of the silent period was greater in VOL than in EVO ($P = 0.002$).

## Discussion

In this study, we investigated the influence of descending drive on the fatigue-induced changes in cortico-motoneuronal excitability of human knee extensors. To this end, cortical and motoneuronal changes induced

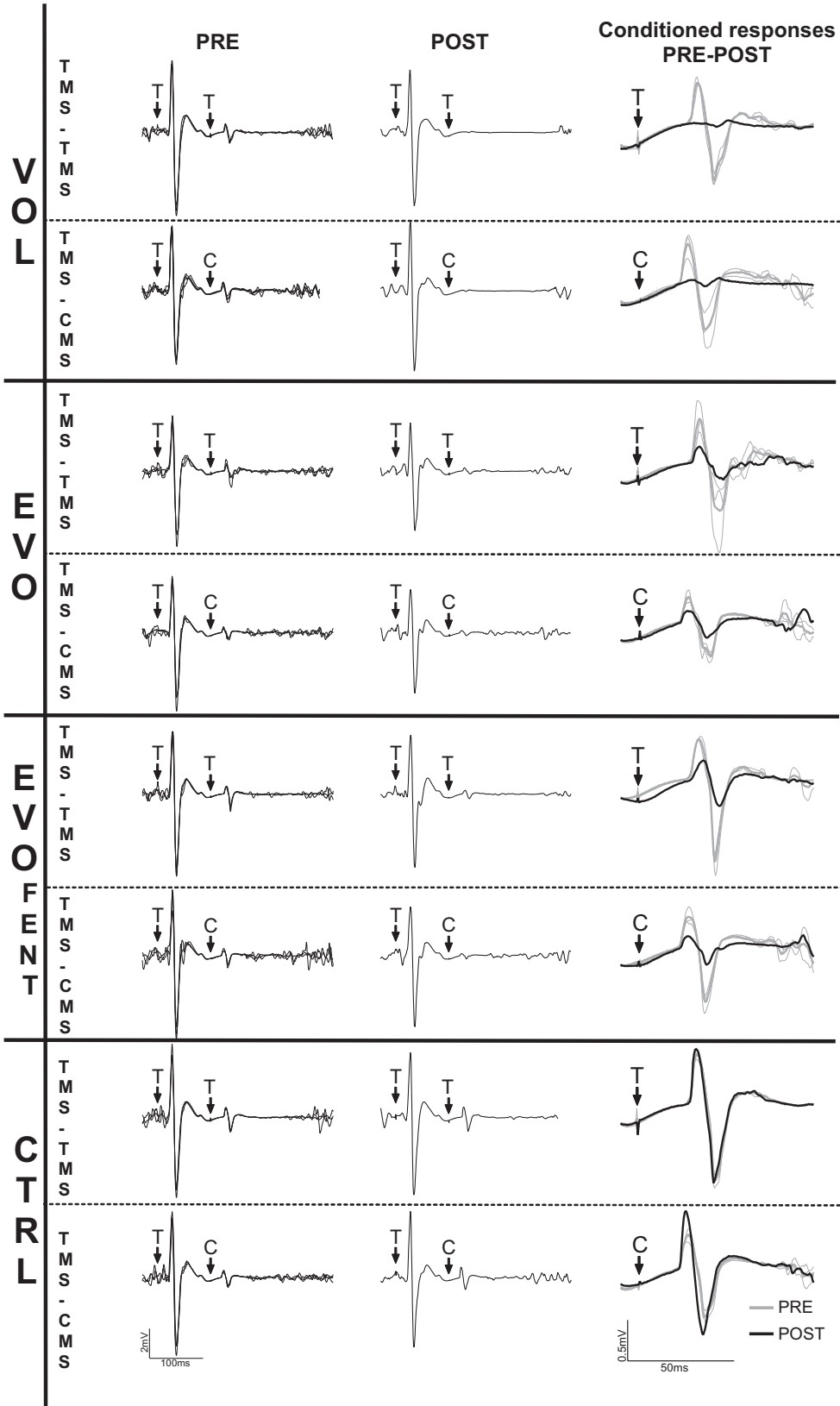

**Figure 2. Representative traces of EMG responses**
Representative traces of the EMG responses to paired stimulations (CMS, C, cervicomedullary electric stimulations; TMS, T, transcranial magnetic stimulations) before (three individual overlaid traces) and after intermittent

isometric quadriceps contractions resulting in a ~40% reduction in twitch torque. The top row within each condition illustrates responses to paired (interstimulus interval, 100 ms) TMS–TMS stimuli, whereas the bottom row illustrates paired TMS–CMS stimuli. The right column depicts pre- to postexercise changes in conditioned transcranial and cervicomedullary motor-evoked potentials in each of the four conditions. Thin lines represent individual pre-exercise traces, and thick lines represent the average of the pre-exercise responses (grey) and the postexercise response (black) used in data analysis. Other abbreviations: CTRL, control protocol; EVO, electrically evoked contractions; EVO$_{FENT}$, electrically evoked contractions plus fentanyl; and VOL, voluntary contractions.

by submaximal voluntary quadriceps contractions were compared with those induced by electrically evoked contractions leading to the same degree of end-exercise peripheral fatigue. Although descending drive accounted fully for the cortical changes associated with voluntary exercise, it explained only ~70% of the reductions in motoneuron excitability. We therefore evaluated the contribution of group III/IV muscle afferents in modulating motoneuron excitability during evoked contractions by comparing the modifications with those induced by the same exercise performed with pharmacologically attenuated group III/IV afferent feedback. Our findings suggest that fatiguing voluntary knee-extensor exercise facilitates motor cortical but inhibits (or disfacilitates) motoneuron excitability, with the combined effect of an overall depression of the

corticomotoneuronal pathway. Mechanisms sensitive to descending drive account for the modulation at the motor cortex and for the majority, but not all, of the reduction in the excitability of the knee-extensor motoneuron pool. Group III/IV muscle afferent feedback does not influence the excitability of either component of the corticomotoneuronal pathway during electrically evoked exercise, that is, in the absence of descending drive.

## Corticomotoneuronal changes induced by voluntary knee-extensor exercise

As reflected by the changes in the conditioned responses, which represent alterations in excitability independent of the facilitatory influence of ongoing motor activity

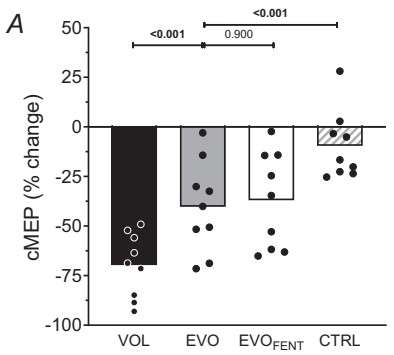 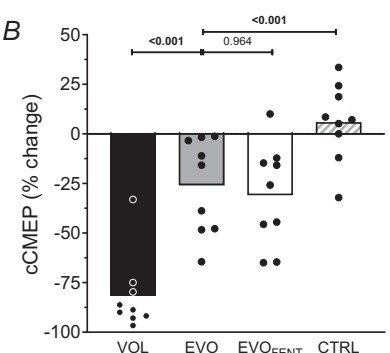 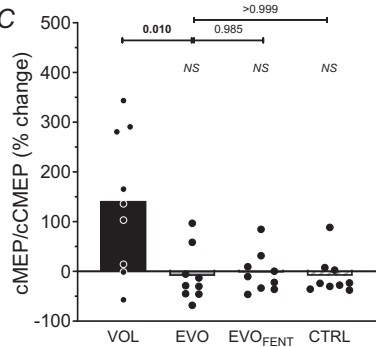

**Figure 3. Conditioned responses**
*Exercise-induced changes in A*, conditioned motor-evoked potentials (cMEP), *B*, conditioned cervicomedullary motor-evoked potentials (cCMEP), and *C*, cMEP/cCMEP ratio. Other abbreviations: CTRL, control protocol; EVO, electrically evoked contractions; EVO$_{FENT}$, electrically evoked contractions plus fentanyl; NS, not significantly different from baseline; and VOL, voluntary contractions.

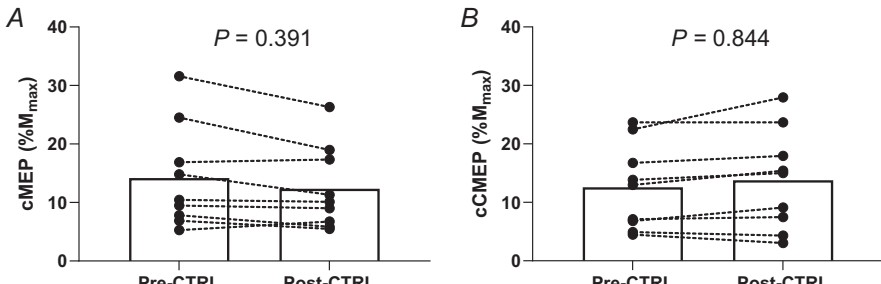

**Figure 4. Pre- to post-exercise changes in conditioned responses for the control protocol**
Individual data of the conditioned motor-evoked potentials (cMEP; *A*) and cervicomedullary motor-evoked potentials (cCMEP; *B*) before and after the control protocol (CTRL), which was matched to the same duration as the electrically evoked contractions (EVO) protocol.

(McNeil et al., 2009), the submaximal voluntary knee-extensor contractions resulted in a decrease in motoneuronal excitability (Figs 2 and 3*B*) and an increase in motor cortical excitability (Figs 2 and 3*C*). Although the former observation corroborates earlier findings based on the same exercise modality (Finn et al., 2018), the data on cortical excitability (when determined in the absence of motor activity) are new. Regardless, the exercise-induced changes in unconditioned responses were different from those in conditioned responses, because both uMEP and uCMEP remained unaltered. The lack of an effect on motoneuron excitability when tested with ongoing motor activity (i.e., uCMEP) was not unexpected and (re-)emphasizes that the excitatory influence associated with motor activity/drive to the motoneuron during the assessment procedure can compensate, and thus mask, fatigue-related motoneuronal depression (Finn et al., 2018; Kennedy et al., 2016; McNeil et al., 2009; Weavil et al., 2015). It should, in this context, be highlighted that because of the fatigue-induced reduction in motoneuron excitability, the magnitude of drive required to maintain the target EMG during the assessment procedure (20% of maximal EMG) was likely to be larger after exercise compared with before. The excitatory influence of drive was therefore stronger during the postexercise assessment and, as reflected by the lack of a pre/postexercise change in uCMEP, fully compensated for the reduction in motoneuron excitability.

Different from the behaviour of conditioned responses (Fig. 3*C*), which indicated increased cortical excitability, uMEP/uCMEP did not change from before to after the exercise. This consistency in unconditioned responses suggests no effect of submaximal, inter-mittent knee-extensor contractions on motor cortical excitability and differs from the previously documented increase following sustained quadriceps MVCs (Kennedy et al., 2016) and the decrease following locomotor exercise (Sidhu et al., 2018). The reason(s) for these divergences remain unknown; however, differences in motor activity/drive between studies during the assessment procedure and the size of the exercising muscle mass might play a role. Specifically, Kennedy et al. (2016) evaluated postexercise excitability during brief MVCs, which require maximal neural drive and thus exert high excitatory influence on the cortex, whereas measurements in the present study were obtained during submaximal contractions characterized by less neural drive and therefore less cortical facilitation. Furthermore, the amount of muscle mass fatiguing during locomotor exercise is substantially larger in comparison to single-joint contractions. This might result in greater group III/IV-mediated inhibitory effects on the motor cortex, leading to a decrease in uMEP/uCMEP during cycling exercise (Sidhu et al., 2018) that is not observed during single-joint contractions (Fig. 3*C*; Kennedy et al.,

2016). Nonetheless, appropriately designed studies are needed to determine the exact mechanisms accounting for the task-dependent effects of exercise on motor cortical excitability. Taken together, although the findings related to unconditioned responses suggest that fatiguing, sub-maximal knee-extensor exercise does not affect cortical or motoneuronal excitability, it is important to consider that this lack of an apparent effect does not necessarily reflect a lack of change in these components of the human motor pathway, but a counterbalance of excitatory and inhibitory influences (Weavil & Amann, 2018). To remove a major excitatory, and thus confounding, influence (i.e., motor activity) from the assessment procedure, we used conditioned responses and found that voluntary submaximal quadriceps exercise compromises motoneuronal excitability and facilitates motor cortical excitability. Given that we were interested primarily in isolating the effect of descending drive and group III/IV muscle afferents on corticomotoneuronal excitability, the remainder of the Discussion focuses on findings related to conditioned responses.

### Role of descending drive and group III/IV muscle afferent feedback in modulating corticomotoneuronal changes during knee-extensor exercise

To test the hypothesis that descending drive influences fatigue-induced alterations in corticomotoneuronal excitability, we compared the changes seen after voluntary knee-extensor contractions with those following electrically evoked contractions causing the same degree of peripheral fatigue (Fig. 1) and therefore the same degree of intramuscular metabolic perturbation and likely group III/IV muscle afferent feedback (Blain et al., 2016). The substantially smaller decrease in cCMEP shortly after EVO compared with VOL suggests that mechanisms sensitive to descending drive account for the majority, but not all, of the fatigue-induced reduction in motoneuron excitability during voluntary knee-extensor exercise (Fig. 3*B*). Data available from this investigation do not allow us to identify the mechanisms by which descending drive inhibits or disfacilitates motoneuron excitability; however, neural drive-related changes in intrinsic motoneuronal properties (Gorman et al., 2005), insufficient release or depletion of neurotransmitters (Zucker & Regehr, 2002) and spinal neuromodulatory systems activated by descending drive (Kavanagh & Taylor, 2022; Khan et al., 2016) might all play a role (Amann et al., 2022). Changes related to repetitive activation are highly likely, given the overlap of the motoneurons used during the fatiguing task and the size of the conditioned responses (Brownstein et al., 2021; Finn et al., 2018; McNeil et al., 2011). Also, descending drive associated with strong (but not weak) muscle contractions

can cause serotonin spillover onto the initial segment of motor axons and can inhibit motoneuron excitability and output (Kavanagh & Taylor, 2022; Thorstensen et al., 2021). In our study, EMG activity during VOL increased from the first contraction to exercise termination (from $18 \pm 6$ to $40 \pm 9\%$ of $MVC_{EMG}$). This level of activation is unlikely to cause spillover; however, it cannot be ruled out (Henderson et al., 2022). Therefore, given that serotonin can either facilitate or inhibit motoneuron excitability depending on contraction intensity (Henderson et al., 2022), its role in explaining the changes observed during VOL in the present study remains speculative.

Furthermore, the observation that cMEP/cCMEP (Fig. 3*C*) and the silent period increased during VOL but not during EVO suggests that mechanisms sensitive to descending motor drive/the repetitive activation of the cortex might simultaneously increase both motor cortical excitability and inhibitory processes along the motor pathway. This is consistent with studies based on elbow flexor contractions (Klass et al., 2008; Lévénez et al., 2008; Søgaard et al., 2006), which also suggest that the contraction-induced rise in cortical excitability is largely independent of Ia- and group III/IV-mediated afferent feedback (Taylor et al., 1996, 2000). Additionally, only VOL resulted in reductions in voluntary activation, providing evidence of a mechanism related to descending drive inhibiting voluntary motor output (D'Amico et al., 2020).

Despite the removal of descending drive from the fatigue protocol, a significant decrease in cCMEPs ($\sim$25%) was still observed following the $\sim$6 min evoked submaximal knee-extensor contractions (i.e., EVO; Fig. 3*B*), and this contrasts with the lack of a change in motoneuron excitability observed after 10 s (Gandevia et al., 1999; Khan et al., 2016) and 2 min (D'Amico et al., 2020) of evoked maximal contractions of upper-limb/hand muscle. The discrepancy between these studies might be explained by differences in exercise protocols (leg *vs.* arm/hand and submaximal *vs.* maximal contractions), assessment methods (CMEPs *vs.* F-waves and conditioned *vs.* unconditioned responses) (McNeil et al., 2013; Thorstensen et al., 2022), and the duration and intensity of the contraction (Khan et al., 2012; McNeil et al., 2009, 2011). Nevertheless, it indicates that descending drive is not required to observe a change in motoneuron excitability.

To investigate group III/IV muscle afferent feedback as a potential mechanism mediating the decrease in motoneuronal excitability following EVO, we compared the changes with those observed during the same exercise performed with blocked feedback from these sensory neurons (i.e., EVO$_{FENT}$). In comparison to EVO, we found no effect of blocking feedback from group III/IV afferents on corticomotoneuronal excitability. This suggests that the reduction in motoneuronal excitability following EVO

occurred independent of feedback from these sensory neurons (Fig. 3*B*). Furthermore, based on the observations during the CTRL protocol, the short MVCs involved in the neuromuscular function assessments conducted every minute throughout each protocol had no influence on the decrease in motoneuron excitability seen during EVO and EVO$_{FENT}$.

Although some of the previous studies based on single-joint exercise suggest that group III/IV muscle afferent feedback has no influence on cortico-motoneuronal excitability (Butler et al., 2003; Hilty et al., 2011; Kennedy et al., 2016; Taylor et al., 1996, 2000), others postulate significant facilitatory and inhibitory effects (Hodges et al., 2021; Martin et al., 2006, 2008; Martinez-Valdes et al., 2020), even during locomotor exercise (Sidhu et al., 2014, 2017, 2018). Accordingly, there is no characteristic response that allows a categorical statement about the influence of group III/IV muscle afferent feedback on motoneuronal and cortical excitability. The ultimate impact is likely to depend on the exercise protocol (flexors *vs.* extensors, lower- *vs.* upper-limb muscles, voluntary *vs.* evoked contractions, small *vs.* large muscle mass), the methodology used to manipulate afferent feedback (augmented feedback via postexercise circulatory occlusion or experimental pain *vs.* pharmacologically attenuated feedback) and the method used to quantify excitability (e.g., conditioned *vs.* unconditioned responses, motor-evoked potentials *vs.* F-waves) (Amann et al., 2022). Considering this specificity, the present findings suggest that when descending drive is transiently suppressed during the silent period (which inhibits cortical output and disfacilitates the motoneurons), feedback from the exercising quadriceps does not contribute to the cortico-motoneuronal changes observed during intermittent isometric fatiguing knee-extensor contractions.

Although the precise mechanism underlying the decrease in cCMEPs following electrical nerve stimulation is unclear, a few factors might contribute. Despite previous studies not implicating this mechanism in a change in motoneuron excitability, the most likely explanation is that repetitive antidromic and reflex activation (Bergquist et al., 2011) altered the intrinsic properties of the involved motoneurons, reducing their likelihood of firing. Furthermore, although speculative, the electrical stimulation could have hyperpolarized axons in both sensory and motor pathways (Brock et al., 1953; Kiernan et al., 1997, 2004). Combined with potential reductions in muscle spindle discharge happening during sustained contractions (Macefield et al., 1991), hyperpolarization of the sensory pathway might have decreased excitatory Ia input further, resulting in motoneuron disfacilitation. If repetitive activation is indeed the mechanism to explain the reduction in EVO in spite of evidence suggesting otherwise (D'Amico et al., 2020; Gandevia et al., 1999;

Khan et al., 2016), then it is possible that the reduction in cCMEP is underestimated in comparison to the entire pool excitability. Submaximal evoked contractions would excite the axons of a random sample of motoneurons (Bergquist et al., 2011; Bickel et al., 2011), including ones that were and were not able to be assessed by the cCMEP, given its size (i.e., $\sim$ >15% of $M_{max}$). However, the absence of a reduction in voluntary activation, which results from involvement of the whole motoneuron pool, following EVO argues against a larger decrease in motoneuron excitability in this condition.

### Experimental considerations and future directions

Although the inclusion of a condition combining voluntary contractions with group III/IV afferent blockade would have provided additional insight, this approach was not feasible within the scope of the present study. Lumbar intrathecal fentanyl administration is a highly invasive procedure that substantially increases participant burden and risk. Given these constraints, and in order to maintain a controlled design that directly addressed our primary research question, we focused on electrically evoked contractions to isolate the effect of group III/IV feedback on corticomotoneuronal excitability. In our previous studies using lumbar intrathecal fentanyl and conditioned knee-extensor MEPs and CMEPs in conjunction with voluntary cycling exercise (i.e., requiring descending drive), group III/IV muscle afferents were found to have no influence on the motoneuron pool but an inhibitory effect on cortical excitability (Sidhu et al., 2018). This impact is different from the lack of an effect during the evoked contractions in the present study and makes it tempting to speculate that the cortical consequence of group III/IV muscle afferent feedback is contingent on its interaction with voluntary drive. Appropriately designed studies are needed to explore this possibility.

Another consideration is the inherent variability of both MEPs and CMEPs, which can reduce sensitivity to detect subtle changes in excitability. The number of assessments in the present study was limited to maintain an acceptable balance between data reliability and participant burden, particularly given the discomfort associated with cervicomedullary stimulations and the short time window of recovery in the postexercise phase. These constraints might have contributed to the variability observed in the measures. To provide a visual representation of the typical variability observed in these measures, we have presented the CTRL condition data before and after the intervention (Fig. 4), along with individual traces from a representative participant (Fig. 2). Importantly, the significant changes observed in the study were relatively large in magnitude, whereas the

non-significant differences were minimal and closely matched between conditions. This pattern makes it unlikely that meaningful effects were either missed or exaggerated owing to variability.

### Conclusion

Mechanisms sensitive to voluntary descending drive account for the increase in motor cortical excitability and contribute, but do not account fully for the decrease in motoneuronal excitability during intermittent isometric fatiguing single-leg knee-extension exercise. Group III/IV leg muscle afferent feedback can be excluded as a mechanism determining the corticomotoneuronal changes during this exercise modality.

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

## Additional information

### Data availability statement

The data of this study are available from the corresponding author upon reasonable request.

### Competing interests

The authors declare no conflict of interest.

### Author contributions

F.G.L., J.C.W., V.P.G. and M.A. contributed to the conception and design of the study. J.C.W., V.P.G., T.S.T., H.-Y.W., N.M.B., J.E.J. and S.R.J. performed the experiments. F.G.L., V.P.G. and H.T.F. analysed the data. F.G.L., J.C.W., V.P.G., H.T.F. and M.A. interpreted the data. F.G.L. drafted the manuscript, and all authors were involved in revising it critically for important intellectual content. All authors have read and approved the final version of the manuscript and agree to be accountable for all aspects of the work in ensuring that questions related to the accuracy or integrity of any part of the work are appropriately investigated and resolved. All persons designated as authors qualify for authorship, and all those who qualify for authorship are listed.

### Funding

This study was supported by the National Heart, Lung, and Blood Institute (HL-162856 and HL-170007) and the U.S. Veterans Affairs Rehabilitation Research and Development (E3343-R).

### Acknowledgements

We thank all participants for their time and effort. We also thank Dr Ryan Broxterman for their valuable advice and guidance on the statistical analyses. The Abstract figure was created with BioRender.

### Keywords

fatigue, motoneuron, motor cortex, neural feedback, voluntary drive

## Supporting information

Additional supporting information can be found online in the Supporting Information section at the end of the HTML view of the article. Supporting information files available:

**Peer Review History**

