## [Peer Review History · The Journal of Physiology]

On the role of descending drive and group III/IV muscle afferent feedback in modulating corticomotoneuronal excitability during knee-extensor exercise

Fabio Giuseppe Laginestra, Joshua C Weavil, Vincent P. Georgescu, Taylor S Thurston, Hsuan-Yu Wan, Nathaniel M Birgenheier, Jacob E. Jessop, Harrison Thomas Finn, Scott R Junkins, and Markus Amann
DOI: 10.1113/JP289021

Corresponding author(s): Fabio Giuseppe Laginestra (fabio.laginestra@utah.edu)

The following individual(s) involved in review of this submission have agreed to reveal their identity: Callum G Brownstein (Referee #1)

Review Timeline:

Submission Date:	05-Apr-2025
Editorial Decision:	08-May-2025
Revision Received:	28-Oct-2025
Editorial Decision:	19-Nov-2025
Revision Received:	19-Nov-2025
Accepted:	22-Nov-2025

Senior Editor: Richard Carson

Reviewing Editor: Jakob Škarabot

Transaction Report:

Dear Dr Laginestra,

Re: JP-RP-2025-289021 **"On the role of descending drive and group III/IV muscle afferent feedback immodulating corticomotoneuronal excitability during knee-extensor exercise"** by Fabio Giuseppe Laginestra, Joshua C Weavil, Vincent P. Georgescu, Taylor S Thurston, Hsuan-Yu Wan, Nathaniel M Birgenheier, Jacob E. Jessop, Scott R Junkins, and Markus Amann

Thank you for submitting your manuscript to The Journal of Physiology. It has been assessed by a Reviewing Editor and by 2 expert referees and we are pleased to tell you that it is potentially acceptable for publication following satisfactory major revision.

REVISION CHECKLIST:

We look forward to receiving your revised submission.

Yours sincerely,

Richard Carson
Senior Editor
The Journal of Physiology

REQUIRED ITEMS

- Author photo and profile. First or joint first authors are asked to provide a short biography (no more than 100 words for one author or 150 words in total for joint first authors) and a portrait photograph. These should be uploaded and clearly labelled together in a Word document with the revised version of the manuscript. See Information for Authors for further details.

- Your manuscript must include a complete Additional Information section, including competing interests; funding; author contributions and acknowledgements.

- Please upload separate high-quality figure files via the submission form.

- Please include an Abstract Figure file, as well as the Figure Legend text within the main article file. The Abstract Figure is a piece of artwork designed to give readers an immediate understanding of the research and should summarise the main conclusions. If possible, the image should be easily 'readable' from left to right or top to bottom. It should show the physiological relevance of the manuscript so readers can assess the importance and content of its findings. Abstract Figures should not merely recapitulate other figures in the manuscript. Please try to keep the diagram as simple as possible and without superfluous information that may distract from the main conclusion(s). Abstract Figures must be provided by authors no later than the revised manuscript stage and should be uploaded as a separate file during online submission labelled as File Type 'Abstract Figure'. Please also ensure that you include the figure legend in the main article file. All Abstract Figures should be created using BioRender. Authors should use The Journal's premium BioRender account to export high-resolution images. Details on how to use and access the premium account are included as part of this email.

EDITOR COMMENTS

Reviewing Editor:

Methods Details:

As per the comments of referees, some minor detail is missing at several points which the authors need to amend/add in their revisions.

Comments to the Author:

This study examined the role of descending drive on corticomotoneuronal excitability during knee extensor exercise. Both reviewers find value in this work, however, they also both outline some concerns that should be considered. In particular,

the authors should justify the lack of group III/IV blockade during voluntary activity and consider whether this influences the robustness of their interpretation. Additionally, there are several points in the manuscript that either require clarification (methods) or careful revisions in interpretation (discussion). Finally, the authors need to justify the use of only 3 responses to stimulations; as they will undoubtedly be aware, these responses are inherently variable. Averaging of responses is not entirely helpful in this regard and the authors should consider other statistical approaches (e.g. mixed modelling) that can account for inherent within-participant variability.

Please also see 'Required Items' above.

REFeree COMMENTS

Referee #1:

General comments

The present study by Laginestra and colleagues investigated the role of descending drive on corticomotoneuronal excitability during knee-extensor exercise. Corticomotoneuronal excitability was assessed using transcranial magnetic and corticomedullary stimulation during submaximal knee extensor exercise. This was performed under voluntary conditions (i.e. in the presence of descending drive) and using electrical nerve stimulation (i.e. in the absence of descending drive). The results demonstrated that reductions in motoneuron excitability (conditioned CMEPs) were largely, but not fully, attenuated in the absence of descending drive. To determine the contribution of group III/IV afferent feedback to the reduced motoneuron excitability during electrically evoked knee extensor exercise, pharmacological blockade of group III/IV afferents was used. This resulted in no difference in the reduction in motoneuron excitability, indicating that group III/IV afferents played no role in contributing to the observed alterations along the corticomotoneuronal pathway.

This is a well conducted and well written study that provides novel insight into the contribution of descending drive to exercise-induced alterations in corticomotoneuronal excitability of locomotor muscles. The methods appear robust, and the conclusions mostly align with the study results. However, I have a few comments and queries which require clarification.

The major gap in this study is that group III/IV blockade was not performed under voluntary conditions. Thus, while the results rule out a contribution of group III/IV afferents to the reduction in motoneuron excitability during involuntary evoked contractions, they do not rule out a possible contribution during voluntary activity. As highlighted by the authors in the discussion, an interaction between group III/IV afferent feedback and voluntary drive remains a possibility, as is indeed indicated by Sidhu et al., 2018. Since this would have required just one additional condition in the present study (i.e. voluntary contractions with fentanyl), why did the authors choose not to include this in the present design?

Methods

Lines 170-173

The authors state that the time delay between TMS and conditioned MEPs was individually chosen to avoid the influence of spinal inhibition on the silent period. However, it is unclear how this was conducted. How did they ensure that there was no influence of spinal inhibition in each individual? Please elaborate on this procedure.

Lines 182-184

Was the intensity of the TMS and CMS delivered in isolation the same as the intensities associated with the conditioning stimulus?

Line 248

The inclusion of a sham condition was a strength of the present study. However, I was surprised to see that this wasn't included in the one-way ANOVA. I would suggest that this is amended for a more appropriate statistical comparison between conditions.

Results

Line 297-299

Looking at the conditioned MEP responses, it appears that the baseline responses were quite substantially lower (around 35%) for the EVOFENT (12.2 {plus minus} 7.6% Mmax) than EVO (18.9 {plus minus} 14.2% Mmax). It would be useful if the authors could run statistics to assess whether there were any differences in baseline response amplitudes. This is of importance given that the reduction in MEP and/or CMEP responses is dependent on their initial amplitude and the contraction intensity of the fatiguing task (e.g. Finn et al., 2018).

I believe that the results section would benefit from confirmation that the target contraction intensity (20% MVC) was matched between the VOL, EVO and EVOFENT conditions, i.e. the average torque throughout the fatiguing protocols. This is not currently reported, and is important to ensure that the conditions were appropriately matched for contraction intensity.

Furthermore, to ensure that EMG was consistent between pre- and post-exercise measurements at met the target level (i.e. 20% iso-EMG), the pre-stimulus EMG should be reported as a % of baseline iso-EMG.

Discussion

Lines 354-357

The authors provide interpretation on the comparison from the results between the present study and that of Kennedy et al. A further comparison of interest here is with that of Sidhu et al (2018, J Physiol) in response to cycling exercise. Despite a similar magnitude of reduction in Q_{tw}, the Sidhu et al study noted a reduction in uMEP/uCMEP, i.e. a reduction in cortical excitability. While the different modalities and intensities might explain these divergent results, I believe some interpretation around this would be helpful, as it might help generate hypotheses on modality-specific differences in neurophysiological adjustments during exercise.

Line 395

The authors state here that mechanisms sensitive to descending motor drive facilitate intracortical inhibition. This is unclear given that, 1) intracortical inhibition was not measured in the present study and, 2) an increase in intracortical inhibition would not be expected to facilitate motor cortical excitability.

Referee #2:

This manuscript reports that vastus lateralis responses to cervicomedullary stimulation elicited during the TMS-evoked silent period are reduced after fatiguing voluntary or stimulated knee extensor contractions but that the reduction is greater after voluntary contractions. In addition, the reduction after stimulated contractions does not depend on group III/IV muscle afferents. MEPs were also evoked during the silent period. They decreased less than CMEPs after voluntary fatiguing contractions but similarly to CMEPs after stimulated contractions with the interpretation that cortical excitability increased after the voluntary exercise and was unchanged by the stimulated contractions.

The major concern with the study is that the findings are based on three baseline responses and one post-contraction response for each potential in each condition for the nine participants. This is a small number, given the variability of these

type of responses. While the responses may be time-sensitive following the contractions, a time course of depression or recovery could have bolstered the findings from immediately after the contractions. At baseline, there would have been no experimental limit on the number of responses collected.

Specific comments

Abstract

Line 48. No definition of conditioned potentials. Also, missing word.

Introduction

Line 74. This sentence is confusing. It is odd to talk about the effect of descending drive on the cortex as it is the origin rather than the recipient of the drive. The sentence also seems to say there is little information on the changes in cortical excitability during voluntary contraction, but there is quite a lot of information about that.

Methods

Line 166. What feedback was given to participants to enable contractions to the target of 20% maximal mean rectified EMG?

Line 181. What was the range of test TMS intensities?

Line 196. What was the duration of the iso-EMG contractions? The term iso-EMG is not defined.

Line 197. Three of each type of potential for baseline values and just one of each after the fatiguing exercise seems too few responses given the variability of MEPs and CMEPs. This needs to be justified.

Line 199. Were the visits for voluntary and evoked contraction protocols in random order?

Line 208. Are these values the mean stimulus intensities at the beginning and end of the fatiguing evoked contraction?

Line 216. Does this mean there is just one post-exercise MEP, CMEP, cMEP and cCMEP?

Line 251-252. Were the paired t-tests for each protocol corrected for multiple comparisons i.e. 4 comparisons for each parameter?

Results

Line 265. As an additional measure of change in neuromuscular function, it would be interesting to know the force produced during the matched EMG contractions performed when the potentials were evoked. Comparison between the voluntary and stimulated condition may give insight into whether the peripheral fatigue involved similar motor units in the two conditions.

Line 311-314. Was the silent period measured after the unconditioned MEP? Or after paired stimuli?

Figure 1. Legend should describe why there are not data points at one min intervals across the time base. It would be helpful to include the data for Sham in this figure even though it was not included in analyses. It should help display the reliability of the measures.

Figure 2. Time scale bar for right panel is incorrect. Latencies are ~twice as long as for left panel.

Are the Pre traces just one selected trace from 3 responses? Or are they averaged? With the limited number of responses, showing all baseline responses overlaid might give some reassurance about their variability.

Figure 3. It would be helpful to see the Sham data to see the variability

Discussion

Line 318. The first sentence is too general to describe the study which is examining the influence of voluntary drive during fatiguing exercise on responses after the fatiguing exercise.

Line 385. The Kiernan et al reference is not entirely appropriate here as it is about changes in axonal excitability primarily with regard to electrical stimulation. It is unknown whether axonal changes have parallel changes at the cell body or axon initial segment to influence action potential generation after synaptic input.

Line 391. The relative strength of the final sustained 20% MVC is likely to be the key to whether there was serotonin spillover during the fatiguing exercise. Was perceived effort (or EMG) high at the end of the sustained contractions?

Line 419-420. While Macefield et al report decreases in muscle spindle firing during sustained submaximal contractions, it is

not obvious how this effect might come about through hyperpolarization of sensory or motor axons (which typically has a minor effect on action potential propagation). It is also not clear why axonal hyperpolarization should suppress cMEPs.

If there are force records during the potentials it might be helpful to check whether alterations in twitch dynamics may have altered afferent feedback at the time of the conditioned potentials i.e. whether the twitch in response to the conditioning TMS is rising (reduced spindle and increased tendon organ firing) or falling (increased spindle firing) and whether this changes with peripheral fatigue.

Line 432. One important factor that is likely to influence findings is whether excitability is measured during contraction or rest. Responses during contraction but conditioned by a TMS-evoked silent period are not the same as those during rest.

Line 433-434. While it is OK to argue that the TMS evoked silent period removes the influence of descending drive on the motoneurons (with the caveat that the motoneuron volley evoked by the conditioning stimulus has sequelae at a spinal level that last for an uncertain period), the silent period has its own inhibitory effects at a cortical level so that it does not just remove voluntary-contraction-related facilitation, it causes inhibition.

END OF COMMENTS

EDITOR COMMENTS

Reviewing Editor:

Methods Details:

As per the comments of referees, some minor detail is missing at several points which the authors need to amend/add in their revisions.

Comments to the Author:

This study examined the role of descending drive on corticomotoneuronal excitability during knee extensor exercise. Both reviewers find value in this work, however, they also both outline some concerns that should be considered. In particular, the authors should justify the lack of group III/IV blockade during voluntary activity and consider whether this influences the robustness of their interpretation. Additionally, there are several points in the manuscript that either require clarification (methods) or careful revisions in interpretation (discussion). Finally, the authors need to justify the use of only 3 responses to stimulations; as they will undoubtedly be aware, these responses are inherently variable. Averaging of responses is not entirely helpful in this regard and the authors should consider other statistical approaches (e.g. mixed modelling) that can account for inherent within-participant variability.

Please also see 'Required Items' above.

Thank you very much for serving as the Reviewing Editor for our manuscript. We found the reviewers' feedback to be highly constructive and helpful in improving the quality of our work. We have addressed all comments, performed the requested analyses, and incorporated the suggested revisions. As a result, we believe the revised manuscript is now more comprehensive and substantially strengthened compared to the original submission.

Please note that we have added Dr. Harrison Finn as a co-author. His contribution to the technical analysis of the data and the statistical aspects of the revision warrants authorship. Furthermore, he provided critical feedback adding nuance and depth to the manuscript.

Please find our responses in blue and revised text of all the changes in red in the manuscript.

REFEREE COMMENTS

Referee #1:

General comments

The present study by Laginestra and colleagues investigated the role of descending drive on corticomotoneuronal excitability during knee-extensor exercise. Corticomotoneuronal excitability was assessed using transcranial magnetic and corticomedullary stimulation during submaximal knee extensor exercise. This was performed under voluntary conditions (i.e. in the presence of descending drive) and using electrical nerve stimulation (i.e. in the absence of descending drive). The results demonstrated that reductions in motoneuron excitability (conditioned CMEPs) were largely, but not fully, attenuated in the absence of descending drive. To determine the contribution of group III/IV afferent feedback to the reduced motoneuron excitability during electrically evoked knee extensor exercise, pharmacological blockade of group III/IV afferents was used. This resulted in no difference in the reduction in motoneuron excitability, indicating that group III/IV afferents played no role in contributing to the observed alterations along the corticomotoneuronal pathway.

This is a well conducted and well written study that provides novel insight into the contribution of descending drive to exercise-induced alterations in corticomotoneuronal excitability of locomotor muscles. The methods appear robust, and the conclusions mostly align with the study results. However, I have a few comments and queries which require clarification.

Response: Thank you for taking the time to review our manuscript and for appreciating the work that went into it. The comments below were very insightful and surely contributed to strengthening the manuscript and improving its overall quality.

The major gap in this study is that group III/IV blockade was not performed under voluntary conditions. Thus, while the results rule out a contribution of group III/IV afferents to the reduction in motoneuron excitability during involuntary evoked contractions, they do not rule out a possible contribution during voluntary activity. As highlighted by the authors in the discussion, an interaction between group III/IV afferent feedback and voluntary drive remains a possibility, as is indeed indicated by Sidhu et al., 2018. Since this would have required just one additional condition in the present study (i.e. voluntary contractions with fentanyl), why did the authors choose not to include this in the present design?

Response: We agree that including a condition with voluntary contractions under group III/IV blockade would have provided valuable additional information. However, this procedure is highly invasive and would have substantially increased both participant burden and risk. Thus, it was not a matter of simply adding another testing session since such an intervention raises additional ethical considerations and participant safety concerns. We reasoned that performing two spinal blocks in healthy participants within the same study would be excessive. Given these constraints, and because we needed to prioritize one approach, we chose what we considered the

cleanest and most controlled design (using electrically-evoked contractions) which would allow us to isolate group III/IV afferent feedback and directly address our primary research question.

As acknowledged in the manuscript, the potential interaction between voluntary descending drive and group III/IV afferent feedback remains an important consideration for future studies. In light of this comment and that of the Reviewing Editor, we have now further emphasized this point by adding a dedicated section (Experimental considerations and future directions) explicitly noting this as a limitation of the present study.

Methods

Lines 170-173

The authors state that the time delay between TMS and conditioned MEPs was individually chosen to avoid the influence of spinal inhibition on the silent period. However, it is unclear how this was conducted. How did they ensure that there was no influence of spinal inhibition in each individual? Please elaborate on this procedure.

Response: We appreciate the reviewer's comment and acknowledge that our original wording was unclear. We have now revised the text accordingly. When selecting the interstimulus interval for the second pulse during the practice visit, we observed in some participants residual activity during the silent period around 100 ms (likely long-latency reflex activity due to muscle spindles). To avoid misinterpreting this activity as a conditioned MEP/CMEP, especially in the post-exercise phase when conditioned responses are sometimes obliterated, we adjusted the interstimulus interval individually and maintained it consistent across all sessions for the same participant. Although it is not possible to completely exclude spinal inhibition, evidence from the cited literature indicates that interstimulus intervals above 80 ms should produce comparable results.

Revised text in Methods (Lines 169-174):

To ensure that the test stimulus was delivered during a period of complete EMG quiescence, the interstimulus interval was determined individually for each participant. Specifically, if residual EMG activity was observed at 100 ms, the interval was adjusted (always >80 ms) until no activity was visible and then maintained consistently across all trials for that participant (mean \pm SD: 93 ± 6 ms). Based on previous literature, interstimulus intervals above 80 ms should produce comparable outcomes while reducing the likelihood of spinal inhibition (Yacyszyn et al., 2016).

Lines 182-184

Was the intensity of the TMS and CMS delivered in isolation the same as the intensities associated with the conditioning stimulus?

Response: They were the same intensity as the "test/conditioned" stimulus. We have now included the % stimulator output for TMS and specified that we used this intensity also for unconditioned MEP responses.

Revised text in Methods (Lines 182-186):

The test stimulus intensity for TMS was then set to elicit a cMEP response that was approximately equivalent in size to the cCMEP response (test TMS intensity = $65 \pm 12\%$ stimulator output). TMS and CMS were also delivered in isolation (i.e., without a conditioning stimulus) and at the same intensity used for test stimuli, with the purpose of evoking unconditioned MEPs (uMEP) and unconditioned CMEPs (uCMEP).

Line 248

The inclusion of a sham condition was a strength of the present study. However, I was surprised to see that this wasn't included in the one-way ANOVA. I would suggest that this is amended for a more appropriate statistical comparison between conditions.

Response: Thanks for this comment. In this version, we have now included this condition in the main statistical analysis and it is also presented in all figures. Moreover, we added a new figure (Figure 4) reflecting individual data and the variability in this condition. Importantly, to address this and the Reviewing Editor's comment, we reanalyzed the data using linear mixed modeling instead of a one-way ANOVA. This approach provides a more robust framework for handling repeated measures and inter-individual variability. The Statistical Analysis section of Methods has been adjusted accordingly. Please note that upon further consideration, we have decided to change the name of this condition from *SHAM* to *CTRL*, as we felt that *SHAM* did not accurately reflect the nature of this protocol.

Results

Line 297-299

Looking at the conditioned MEP responses, it appears that the baseline responses were quite substantially lower (around 35%) for the EVOFENT ($12.2 \pm 7.6\%$ Mmax) than EVO ($18.9 \pm 14.2\%$ Mmax). It would be useful if the authors could run statistics to assess whether there were any differences in baseline response amplitudes. This is of importance given that the reduction in MEP and/or CMEP responses is dependent on their initial amplitude and the contraction intensity of the fatiguing task (e.g. Finn et al., 2018).

Response: The reviewer raises a good point. To address this, before interpreting the exercise-induced changes in cMEPs and cCMEPs we now compared baseline values for these two indices. After going back and analyzing them at baseline against each other, we found that although the responses look slightly different, there were no differences in cMEP (all $P \geq 0.088$ between conditions), or cCMEP (all $P \geq 0.514$). These comparisons have now been added to the Results section.

Revised text in Methods (Lines 265-266):

Baseline cMEP and cCMEP size expressed as $\%M_{\max}$ were also compared.

Revised text in Results (Lines 341-342 & Line 348):

At baseline, cMEPs were not different between experimental conditions (all $P \geq 0.088$).

At baseline, cMEPs were not different between experimental conditions (all $P \geq 0.514$).

I believe that the results section would benefit from confirmation that the target contraction intensity (20% MVC) was matched between the VOL, EVO and EVOFENT conditions, i.e. the average torque throughout the fatiguing protocols. This is not currently reported, and is important to ensure that the conditions were appropriately matched for contraction intensity.

Furthermore, to ensure that EMG was consistent between pre- and post-exercise measurements at met the target level (i.e. 20% iso-EMG), the pre-stimulus EMG should be reported as a % of baseline iso-EMG.

Response: We agree and now include the requested information in the manuscript. Specific to the EMG data, we have averaged all the pre-stimuli EMG (200 ms window for uMEP, uCMEP, cMEP, and cCMEP) in a single value, and statistically tested it versus the same value for baseline and across conditions.

Revised text in Methods (Lines 252-254):

To ensure that EMG activity was comparable between baseline and post-exercise testing contractions, the root mean square over the 200 ms EMG before every stimulation was analyzed and averaged.

Revised text in Results (Lines 289-290 & Lines 313-318):

The average torque produced during the fatiguing contractions was not different between conditions (VOL: 20.3 ± 3.4 %MVC, EVO: 20.5 ± 3.2 %MVC, EVO_{FENT}: 20.1 ± 4.2 %MVC, $P = 0.887$).

The background EMG activity before stimulation did not change from pre- to post-exercise (VOL: 0.18 ± 0.06 to 0.19 ± 0.07 mV, $B = 6\%$, $P = 0.673$, CI95 [-4,15]; EVO: 0.18 ± 0.06 to 0.19 ± 0.07 mV, $B = -4\%$, $P = 0.558$, CI95 [-2,11]; EVO_{FENT}: 0.18 ± 0.08 to 0.17 ± 0.08 mV, $B = -2\%$, $P = 0.982$, CI95 [-10,6]; CTRL: 0.18 ± 0.07 to 0.17 ± 0.07 mV, $B = -1\%$, $P = 0.932$, CI95 [-4,2]). Also, EMG activity expressed as % pre was not different between conditions (VOL: $106 \pm 14\%$, EVO: $104 \pm 10\%$, EVO_{FENT}: $98 \pm 12\%$, CTRL: $99 \pm 4\%$, all $P \geq 0.356$).

Discussion

Lines 354-357

The authors provide interpretation on the comparison from the results between the present study and that of Kennedy et al. A further comparison of interest here is with that of Sidhu et al (2018, J Physiol) in response to cycling exercise. Despite a similar magnitude of reduction in Q_{tw}, the Sidhu et al study noted a reduction in uMEP/uCMEP, i.e. a reduction in cortical excitability. While the different modalities and intensities might explain these divergent results, I believe some interpretation around this would be helpful, as it might help generate hypotheses on modality-specific differences in neurophysiological adjustments during exercise.

Response: We agree with the reviewer and edited the manuscript accordingly, by adding the Sidhu reference and relative discussion around it in that section

Revised text in Discussion (Lines 400-415):

Different from the behavior of conditioned responses (**Figure 3C**), which indicated increased cortical excitability, uMEP/uCMEP did not change from before to after the exercise. This consistency in unconditioned responses suggests no effect of submaximal, intermittent knee-extensor contractions on motor cortical excitability and differs from the previously documented increase following sustained quadriceps MVCs (Kennedy *et al.*, 2016) and the decrease following locomotor exercise (Sidhu *et al.*, 2018). The reason(s) for these divergences remain unknown, but between study differences in motor activity/drive during the assessment procedure and the size of the exercising muscle mass might play a role. Specifically, while Kennedy *et al.* (2016) evaluated post-exercise excitability during brief MVCs, which require maximal neural drive and thus exert high excitatory influence on the cortex, measurements in the current study were obtained during submaximal contractions characterized by less neural drive and therefore less cortical facilitation. Furthermore, the amount of muscle mass fatiguing during locomotor exercise is substantially larger compared to single-joint contractions. This might result in greater group III/IV-mediated inhibitory effects on the motor cortex, leading to a decrease in uMEP/uCMEP during cycling exercise (Sidhu *et al.*, 2018) that is not observed during single-joint contractions (**Figure 3C**; (Kennedy *et al.*, 2016). Nonetheless, appropriately designed studies are needed to determine the exact mechanisms accounting for the task-dependent effects of exercise on motor cortical excitability.

Line 395

The authors state here that mechanisms sensitive to descending motor drive facilitate intracortical inhibition. This is unclear given that, 1) intracortical inhibition was not measured in the present study and, 2) an increase in intracortical inhibition would not be expected to facilitate motor cortical excitability.

Response: Thank you for pointing that out. We have now revised this initially confusing statement.

Revised text in Discussion (Lines 450-453):

Furthermore, the observation that cMEP/cCMEP (**Figure 3C**) and the silent period increased during VOL but not during EVO suggests that mechanisms sensitive to descending motor drive / the repetitive activation of the cortex may simultaneously increase both motor cortical excitability and inhibitory processes along the motor pathway.

Referee #2:

This manuscript reports that vastus lateralis responses to cervicomedullary stimulation elicited during the TMS-evoked silent period are reduced after fatiguing voluntary or stimulated knee extensor contractions but that the reduction is greater after voluntary contractions. In addition, the reduction after stimulated contractions does not depend on group III/IV muscle afferents. MEPs were also evoked during the silent period. They decreased less than CMEPs after voluntary fatiguing contractions but similarly to CMEPs after stimulated contractions with the interpretation that cortical excitability increased after the voluntary exercise and was unchanged by the stimulated contractions.

The major concern with the study is that the findings are based on three baseline responses and one post-contraction response for each potential in each condition for the nine participants. This is a small number, given the variability of these type of responses. While the responses may be time-sensitive following the contractions, a time course of depression or recovery could have bolstered the findings from immediately after the contractions. At baseline, there would have been no experimental limit on the number of responses collected.

Thank you for the very thorough and thoughtful review of our paper. We believe that the extra analyses you asked for led to additional insights strengthening the overall quality of the paper. We also recognize that some of the points raised highlight important limitations of the present study, particularly regarding the limited number and variable nature of the responses. We therefore added an Experimental Considerations section to acknowledge these issues. Further, we have revised the manuscript by doing everything possible to present the data transparently and to illustrate the inherent variability in our results. We hope these additions provide reassurance about the robustness of our findings despite these limitations.

Specific comments

Abstract

Line 48. No definition of conditioned potentials. Also, missing word.

Response: Thank you for catching these errors. We have now corrected them.

Revised text in Abstract (Lines 46-49):

During constant-EMG contractions before and immediately after exercise, transcranial magnetic (TMS) and cervicomedullary stimulations were administered to elicit conditioned (preceded by a conditioning TMS-pulse) *vastus lateralis* motor-evoked (cMEP) and cervicomedullary motor-evoked (cCMEP) potentials.

Introduction

Line 74. This sentence is confusing. It is odd to talk about the effect of descending drive on the cortex as it is the origin rather than the recipient of the drive. The sentence also seems to say there is little information on the changes in cortical excitability during voluntary contraction, but there is quite a lot of information about that.

Response: The reviewer raises a good point. We now reworded this sentence for clarity. We agree that the cortex is the origin of the descending drive, but we wanted to convey that during exercise the cortex itself is recipient of inputs that can change its excitability when an external stimulus is applied.

Also, while we agree that studies have investigated changes in cortical excitability during voluntary exercise (we cite several throughout the manuscript, later in the discussion), studies specifically linking cortical changes to descending drive are scarce. Observing changes in cortical excitability during voluntary exercise could still be explained by other factors and does not allow to make strong inferences about the role of descending drive, unless specific designs are used.

Revised text in Introduction (Lines 74-75):

Despite little available data on their role on cortical excitability (Khaslavskaja & Sinkjaer, 2005; Luft et al., 2005), factors associated with descending drive can compromise the excitability of motoneurons during fatiguing muscle contractions

Methods

Line 166. What feedback was given to participants to enable contractions to the target of 20% maximal mean rectified EMG?

Response: After performing the MVCs at the beginning of each session, the filtered EMG signal was rectified and smoothed, and the maximal value identified. Using our custom Spike scripts, participants were then provided with real-time visual feedback displaying the target value (20% of the maximal rectified EMG) and a live indicator reflecting the ongoing EMG activity during contraction. Participants were instructed to maintain the contraction by matching the live indicator to the target line. We have added some information to the manuscript for completeness.

Revised text in Methods (Lines 163-167):

For the determination of stimulation intensities and during the successive TMS and CMS procedures, all participants were instructed to perform quadriceps contractions corresponding to 20% of the maximum EMG activity recorded during MVCs. Real-time visual feedback based on the rectified and smoothed (500 ms window) EMG was provided to guide subject in maintaining the target level.

Line 181. What was the range of test TMS intensities?

Response: This information has now been added to the manuscript.

Revised text in Methods (Lines 183-185):

The test stimulus intensity for TMS was then set to elicit a cMEP response that was

approximately equivalent in size to the cCMEP response (test TMS intensity = $65 \pm 12\%$ stimulator output).

Line 196. What was the duration of the iso-EMG contractions? The term iso-EMG is not defined.

Response: We have now clarified this detail in the current version. We were aiming for 15 s with one stimulation administered every 3 s. However, we would still look at the live EMG feedback to make sure it was delivered when subjects resumed a stable 20% EMG - so sometimes contractions could last a little longer. Also, since the term iso-EMG was not used in any other parts of the manuscript, we eliminated it. It was meant to mean that EMG was matched before and after exercise.

Revised text in Methods (Lines 199-202):

Immediately after, excitability measures (i.e. cCMEP, uCMEP, cMEP, uMEP, M_{\max}) were performed during a ~15-20-s contraction corresponding to 20% of the maximal EMG quantified during previous MVC. The order of the stimulations was randomized during each set and delivered every ~3-5 s (Sidhu *et al.*, 2012).

Line 197. Three of each type of potential for baseline values and just one of each after the fatiguing exercise seems too few responses given the variability of MEPs and CMEPs. This needs to be justified.

Response: This is a valid point and we are aware and agree that both MEPs and CMEPs can exhibit considerable variability. There is no scientific justification that can be offered in response to your comment. Rather, the number of assessments was limited to maintain an acceptable balance between data reliability and participant burden, particularly given the discomfort associated with CMEPs. To acknowledge the constraints associated with the limited number of stimulations, we now address this issue in the newly added Limitations section. To graphically illustrate and statistically consider the variability characteristic for these measures, we now offer revised/new figures (Figures 1-3) and included the CTRL condition in the main analysis. Furthermore, we added one new figure (Figure 4) to show the individual variability in cMEP and cCMEP in our CTRL condition. Please note that upon further consideration, we have decided to change the name of our fourth condition from *SHAM* to *CTRL*, as we felt that *SHAM* did not accurately reflect the nature of this protocol.

Line 199. Were the visits for voluntary and evoked contraction protocols in random order?

Response: The conditions were not randomized. This decision was based on the fact that some participants struggle, or fail, to achieve a 40% reduction in Q_{tw} during voluntary exercise, i.e. our a priori target level of fatigue. This issue has not only been documented in the literature (e.g., Ducrocq *et al.* 2023, MSSE, PMID: 33148973), but was also apparent during our pilot work. In case a subject was unable to voluntarily exercise until our target level of fatigue was achieved, we would have adjusted their subsequent EVO trials accordingly, (i.e. stop the

electrically-evoked exercise when the level of fatigue during their VOL was reached). However, this was not necessary in this cohort since every participant was able to reach 40% decrease in quadriceps twitch torque.

Revised text in Methods (Lines 209-211):

The order of conditions was not randomized to ensure that participants could achieve the target level of fatigue in the VOL condition first, which is typically the most demanding, as some participants may be unable to reach 40% reduction in Q_{tw} .

Line 208. Are these values the mean stimulus intensities at the beginning and end of the fatiguing evoked contraction?

Response: Yes. We now changed the wording to better clarify this.

Revised text in Methods (Lines 214-215):

Stimulation intensity during EVO was manually adjusted to assure that the target torque was maintained throughout the protocol (start of protocol: 27 ± 6 mA; end of protocol: 49 ± 15 mA).

Line 216. Does this mean there is just one post-exercise MEP, CMEP, cMEP and cCMEP?

Response: Yes. In the post-exercise phase, it is challenging to obtain multiple responses because the time course of recovery in these variables is uncertain. Given the evidence that excitability measures can recover rather quickly after exercise cessation (for example Figure 5 in Finn et al. 2018), and considering the time required to administer the different types of stimulation (the matched EMG contraction lasted approximately 15–20 s), we were not confident that averaging multiple responses would accurately represent the post-exercise state. We acknowledge this limitation in the newly added Experimental Considerations section.

Also, in line with the reviewers' and Reviewing Editor's feedback, we have now expanded Figure 2 to show all pre- and post-exercise responses for each participant, added the CTRL protocol to both the figures and the main analysis, and included CTRL pre–post data to better illustrate the variability in our measurements.

Line 251-252. Were the paired t-tests for each protocol corrected for multiple comparisons i.e. 4 comparisons for each parameter?

Response: Given the comments from the Reviewing Editor and Reviewer 1, we have now changed our statistical approach from a one-way ANOVA and paired t-tests to linear mixed models, in which we now encompass both between- and within-conditions comparisons. We confirm that all P-values present in this revised version of the manuscript are corrected following the Sidak procedure.

Results

Line 265. As an additional measure of change in neuromuscular function, it would be interesting to know the force produced during the matched EMG contractions performed when the potentials were evoked. Comparison between the voluntary and stimulated condition may give insight into whether the peripheral fatigue involved similar motor units in the two conditions.

Response: Thank you for this very insightful suggestion. We analyzed the data as requested and now present this in the revised version of the manuscript. We measured the torque during the 500-ms preceding each stimulation and then averaged all these results to give a single value for torque for that condition before and after exercise. We then compared the pre/post changes between conditions. Our results show that there were no differences between the fatiguing protocols in the torque associated with the 20% EMG contraction.

Revised text in Results (Lines 319-323):

The torque associated with the 20% EMG contractions decreased similarly from pre- to post-exercise in all fatiguing protocols (VOL: 59 ± 17 to 42 ± 10 Nm, $B = 28\%$, $P < 0.001$, CI95 [-35,-21]; EVO: 61 ± 16 to 48 ± 14 Nm, $B = -21\%$, $P < 0.001$, CI95 [-29,-14]; EVO_{FENT}: 61 ± 15 to 49 ± 16 Nm, $B = -22\%$, $P < 0.001$, CI95 [-28,-16]), but not in CTRL (58 ± 13 to 57 ± 13 Nm, $B = -2\%$, $P = 0.810$, CI95 [-6,2]).

Line 311-314. Was the silent period measured after the unconditioned MEP? Or after paired stimuli?

Response: The silent period was measured after the unconditioned MEP. We now specify this in the text.

Revised text in Methods (Lines 248-249):

The duration of the silent period was considered as the time interval from the TMS pulse eliciting uMEP to the return of the voluntary EMG (Sidhu *et al.*, 2018).

Figure 1. Legend should describe why there are not data points at one min intervals across the time base. It would be helpful to include the data for Sham in this figure even though it was not included in analyses. It should help display the reliability of the measures.

Response: We have now added further explanation to our data presentation. Furthermore, the CTRL data are now presented in all figures and panels and, in response to the suggestion from Reviewer 1, CTRL data are now also included in the main analysis.

Revised text in Figure 1 caption:

Exercise-induced changes in potentiated quadriceps twitch torque (Q_{tw} , panels A and D), maximal voluntary contraction (MVC, panels B and E) torque, and voluntary activation (Panels C and F). Top row: progressive changes throughout each trial. Bottom row: pre- to post-exercise changes. In the top row, data are shown for the first 3 minutes, during which all participants had data available (one point per minute). The final point represents the mean value at each participant's individual task termination time (\pm SD across participants). In panels A, B, and C,

the first 3 time points are staggered to improve clarity of the graphical illustration. VOL: voluntary contractions; EVO, electrically-evoked contractions; EVO_{FENT}, electrically-evoked contractions + fentanyl; CTRL, control protocol. NS = not significantly different from baseline.

Figure 2. Time scale bar for right panel is incorrect. Latencies are ~twice as long as for left panel.

Response: Thank you very much for catching this error. The time scale has now been fixed.

Are the Pre traces just one selected trace from 3 responses? Or are they averaged? With the limited number of responses, showing all baseline responses overlaid might give some reassurance about their variability.

Response: That is a great suggestion. It was indeed only 1 representative tracing. We have now amended Figure 2 to reflect all the responses for that participant.

Figure 3. It would be helpful to see the Sham data to see the variability

Response: We have now included the CTRL data in Figure 1,2, and 3. Furthermore, we now also offer the CTRL data in a separate figure. Since sometimes %change of small responses can be somewhat deceiving, this allows us to display the variability even more clearly.

Discussion

Line 318. The first sentence is too general to describe the study which is examining the influence of voluntary drive during fatiguing exercise on responses after the fatiguing exercise.

Response: Agreed. We revised the sentence accordingly.

Revised text in Discussion (Lines 367-368):

This study investigated the influence of descending drive on the fatigue-induced changes in corticomotoneuronal excitability of human knee-extensors.

Line 385. The Kiernan et al reference is not entirely appropriate here as it is about changes in axonal excitability primarily with regard to electrical stimulation. It is unknown whether axonal changes have parallel changes at the cell body or axon initial segment to influence action potential generation after synaptic input.

Response: Thank you for catching this error. The reference has been substituted for the appropriate one (Gorman et al. 1985).

Revised text in Discussion (Lines 436-438):

Data available from this investigation do not allow us to identify the mechanisms by which descending drive inhibits, or disfacilitates, motoneuron excitability, however, neural drive-related changes in intrinsic motoneuronal properties (Gorman *et al.*, 2005).

Line 391. The relative strength of the final sustained 20% MVC is likely to be the key to whether there was serotonin spillover during the fatiguing exercise. Was perceived effort (or EMG) high at the end of the sustained contractions?

Response: Thank you for this insightful comment. Although we did not obtain measures of perceived effort, we analyzed the EMG data during the voluntary fatiguing task. We found that EMG went from $18 \pm 6\%$ to $40 \pm 9\%$ of MVC_{EMG} . We have now included this in the Methods, Results and Discussion section. Although we believe it is still difficult to discern with certainty whether an $EMG = 40\% MVC_{EMG}$ is “strong enough” to cause spillover, we exert caution in our speculation. We believe that this is very valuable information for the readers and for future investigations.

Revised text in Methods (Lines 257-259):

To determine the exercise-induced increase in EMG during VOL, EMG activity was quantified as the root mean square of the first 20 s of the first fatiguing contraction and for the last 20 s of each successive contraction until task cessation.

Revised text in Results (Lines 323-324):

EMG activity during VOL increased from $18 \pm 6\%$ to $40 \pm 9\%$ of MVC_{EMG} ($P < 0.001$) from the first contraction to exercise cessation.

Revised text in Discussion (Lines 445-450):

In our study, EMG activity during VOL increased from the first contraction to exercise termination ($18 \pm 6\%$ to $40 \pm 9\%$ of MVC_{EMG}). This level of activation is unlikely to cause spillover; however, it cannot be ruled out (Henderson *et al.*, 2022). Therefore, given that serotonin can either facilitate or inhibit motoneuron excitability depending on contraction intensity (Henderson *et al.*, 2022), its role in explaining the changes observed during VOL in the present study remains speculative.

Line 419-420. While Macefield et al report decreases in muscle spindle firing during sustained submaximal contractions, it is not obvious how this effect might come about through hyperpolarization of sensory or motor axons (which typically has a minor effect on action potential propagation). It is also not clear why axonal hyperpolarization should suppress cMEPs.

Response: We agree that this paragraph was confusing, we have now revised this section. The observation that motoneuron excitability decreased with EVO entirely independent of group III/IV afferent feedback was surprising. Speculating about potential mechanisms has been challenging. Our reasoning for axonal hyperpolarization to suppress cMEPs was that if axonal hyperpolarization indeed happens with electrical stimulation of the sensory axons (as shown by the Kiernan papers), excitatory Ia input to the motoneuron could be less post- compared to pre-exercise (i.e., motoneuron disfacilitation following fatiguing exercise). We have now revised this section by putting more weight on the direct role of the repetitive activation of the motoneuron, and deemphasized this as a speculation. We also further expanded the discussion about repetitive activation adding some more important considerations that enriched the paper.

Revised text in Discussion (Lines 494-510):

Although the precise mechanism underlying the decrease in cCMEPs following electrical nerve stimulation is unclear, a few factors may contribute. Despite previous studies not implicating this mechanism in a change in motoneuron excitability, the most likely explanation is that repetitive antidromic and reflex activation (Bergquist *et al.*, 2011) altered the intrinsic properties of the involved motoneurons, reducing their likelihood of firing. Furthermore, although speculative, the electrical stimulation could have hyperpolarized axons in both sensory and motor pathways (Brock *et al.*, 1953; Kiernan *et al.*, 1997; Kiernan *et al.*, 2004). Combined with potential reductions in muscle spindle discharge happening during sustained contractions (Macefield *et al.*, 1991), hyperpolarization of the sensory pathway may have further decreased excitatory Ia input, resulting in motoneuron disfacilitation. If repetitive activation is indeed the mechanism to explain the reduction in EVO in spite of evidence suggesting otherwise (Gandevia *et al.*, 1999; Khan *et al.*, 2016; D'Amico *et al.*, 2020), then it is possible that the reduction in cCMEP is underestimated in comparison to the entire pool excitability. Submaximal evoked contractions would excite the axons of a random sampling of motoneurons (Bergquist *et al.*, 2011; Bickel *et al.*, 2011), including ones that were and were not able to be assessed by the cCMEP given its size (*i.e.*, $\sim >15\% M_{\max}$). However, the absence of a reduction in voluntary activation, which results from involvement of the whole motoneuron pool, following EVO argues against a larger decrease in motoneuron excitability in this condition.

If there are force records during the potentials it might be helpful to check whether alterations in twitch dynamics may have altered afferent feedback at the time of the conditioned potentials *i.e.* whether the twitch in response to the conditioning TMS is rising (reduced spindle and increased tendon organ firing) or falling (increased spindle firing) and whether this changes with peripheral fatigue.

Response: This is an interesting idea. We retrieved the force recordings during the potential and analyzed the time to peak for the conditioning stimuli, for both cMEPs and cCMEPs. We then compared this to the interstimulus interval (ISI). If time to peak was higher than ISI, it meant that the conditioned response would be delivered during the ascending phase of the twitch, conversely, on the falling phase. Our results showed that 8/9 participants received the conditioned stimulus during the ascending phase of the twitch, while 1 participant during the descending phase. This behavior did not change with peripheral fatigue.

Line 432. One important factor that is likely to influence findings is whether excitability is measured during contraction or rest. Responses during contraction but conditioned by a TMS-evoked silent period are not the same as those during rest.

Response: Although we completely agree with this important distinction, we believe that incorporating it into this specific section of the manuscript is not ideal. In this paragraph, we aimed to highlight the diversity of paradigms and methodological approaches specifically used in studies investigating group III/IV afferents, each of which was directly relevant to the studies cited. The distinction between measurements obtained during contraction versus rest—while

undoubtedly influential does not directly relate to the particular paradigms discussed in this section and is therefore difficult to integrate here.

Line 433-434. While it is OK to argue that the TMS evoked silent period removes the influence of descending drive on the motoneurons (with the caveat that the motoneuron volley evoked by the conditioning stimulus has sequelae at a spinal level that last for an uncertain period), the silent period has its own inhibitory effects at a cortical level so that it does not just remove voluntary-contraction-related facilitation, it causes inhibition.

Response: That is a fair point. We have reworded the sentence by incorporating the reviewer's comment.

Revised text in Discussion (Lines 490-493):

Considering this specificity, the current findings suggest that when the (confounding) influence of descending drive, which facilitates the cortex and the motoneuron, is removed during the silent period (which also actively causes inhibition), feedback from the exercising quadriceps has no influence on the corticomotoneuronal changes seen during intermittent isometric fatiguing knee-extensor contractions.

Dear Dr Laginestra,

Re: JP-RP-2025-289021R1 "**On the role of descending drive and group III/IV muscle afferent feedback immodulating corticomotoneuronal excitability during knee-extensor exercise**" by Fabio Giuseppe Laginestra, Joshua C Weavil, Vincent P. Georgescu, Taylor S Thurston, Hsuan-Yu Wan, Nathaniel M Birgenheier, Jacob E. Jessop, Harrison Thomas Finn, Scott R Junkins, and Markus Amann

Thank you for submitting your manuscript to The Journal of Physiology. It has been assessed by a Reviewing Editor and by 2 expert referees and we are pleased to tell you that it is acceptable for publication following satisfactory revision.

REVISION CHECKLIST:

Please upload two versions of your manuscript text: one with all relevant changes highlighted and one clean version with no changes tracked. The manuscript file should include all tables and figure legends, but each figure/graph should be uploaded as separate, high-resolution files. The journal is now integrated with Wiley's Image Checking service. For further details,

see: <https://www.wiley.com/en-us/network/publishing/research-publishing/trending-stories/upholding-image-integrity-wileys-image-screening-service>

We look forward to receiving your revised submission.

Yours sincerely,

Richard Carson
Senior Editor
The Journal of Physiology

EDITOR COMMENTS

Reviewing Editor:

Thank you to the authors for thoroughly responding to the comments. However, some minor comments remain for the authors to consider.

REFEREE COMMENTS

Referee #1:

Thank you to the authors for the rebuttal and amendments to the manuscript, which I hope they found useful and I believe have improved clarity. I have no further comments.

Referee #2:

The manuscript has been improved by revision and I have only three further comments.

Line 316. Background EMG results for EVO. "EVO: 0.18 {plus minus} 0.06 to 0.19 {plus minus} 0.07 mV, B 315 = -4%, P = 0.558, CI95 [-2,11];" Although not impossible, a negative percentage change with a positive change in means is odd and should be checked, especially as it also matches poorly with the 95% CI.

Line 321. Typo - "= 28%" should be negative i.e. "= -28%"

Line 490-493. Please revise this sentence. The current revised text is poorly expressed because it confounds cortical and motoneuronal excitability, describes descending drive as facilitating cortical neurons when the cortex is the origin and not the target of the drive, and does not specify the effects of the silent period on the cortex versus the motoneurons. The silent period reflects inhibition in the cortex with consequent reduction of descending drive and disfacilitation of the motoneurons.

END OF COMMENTS

EDITOR COMMENTS

Reviewing Editor:

Thank you to the authors for thoroughly responding to the comments. However, some minor comments remain for the authors to consider.

Response: We thank you again for the evaluation of our manuscript and revisions. The additional corrections and suggestions have now been incorporated in this version of the manuscript.

REFEREE COMMENTS

Referee #1:

Thank you to the authors for the rebuttal and amendments to the manuscript, which I hope they found useful and I believe have improved clarity. I have no further comments.

Response: Thank you again for your thoughtful feedback throughout the review process. Your revision contributed to improving the quality of the manuscript.

Referee #2:

The manuscript has been improved by revision and I have only three further comments.

Response: Thank you again for the constructive insights provided during the review process. The remaining comments have now been incorporated in this version of the manuscript.

Line 316. Background EMG results for EVO. "EVO: 0.18 {plus minus} 0.06 to 0.19 {plus minus}

0.07 mV, B 315 = -4%, P = 0.558, CI95 [-2,11];" Although not impossible, a negative percentage change with a positive change in means is odd and should be checked, especially as it also matches poorly with the 95% CI.

Line 321. Typo - "= 28%" should be negative i.e. "= -28%"

Response: Thank you for catching both these errors. You are right, the sign in Line 316 is indeed positive, while the one in Line 321 was negative. These have been now amended in this version.

Line 490-493. Please revise this sentence. The current revised text is poorly expressed because it confounds cortical and motoneuronal excitability, describes descending drive as facilitating cortical neurons when the cortex is the origin and not the target of the drive, and does not specify the effects of the silent period on the cortex versus the motoneurons. The silent period reflects inhibition in the cortex with consequent reduction of descending drive and disfacilitation of the motoneurons.

Response: We agree that the original sentence was somewhat convoluted with some unnecessary and redundant wording. We have streamlined the text now and made it more straight to the point.

Revised text in Discussion (Lines 490-493):

Considering this specificity, the current findings suggest that when descending drive is transiently suppressed during the silent period (which inhibits cortical output and disfacilitates the motoneurons), feedback from the exercising quadriceps does not contribute to the corticomotoneuronal changes observed during intermittent isometric fatiguing knee-extensor contractions.

Dear Dr Laginestra,

Re: JP-RP-2025-289021R2 "**On the role of descending drive and group III/IV muscle afferent feedback immodulating corticomotoneuronal excitability during knee-extensor exercise**" by Fabio Giuseppe Laginestra, Joshua C Weavil, Vincent P. Georgescu, Taylor S Thurston, Hsuan-Yu Wan, Nathaniel M Birgenheier, Jacob E. Jessop, Harrison Thomas Finn, Scott R Junkins, and Markus Amann

We are pleased to tell you that your paper has been accepted for publication in The Journal of Physiology.

Yours sincerely,

Richard Carson
Senior Editor
The Journal of Physiology

IMPORTANT POINTS TO NOTE FOLLOWING ACCEPTANCE OF YOUR PAPER:

- **IMPORTANT NOTICE ABOUT OPEN ACCESS:** To assist authors whose funding agencies mandate immediate public access to published research findings, The Journal of Physiology allows authors to pay an Open Access (OA) fee to have their papers made freely available immediately on publication.

- You can help your research get the attention it deserves! Check out Wiley's free Promotion Guide for best-practice recommendations for promoting your work at: www.wileyauthors.com/eoo/guide. You can learn more about Wiley Editing Services which offers professional video, design, and writing services to create shareable video abstracts, infographics, conference posters, lay summaries, and research news stories for your research at: www.wileyauthors.com/eoo/promotion.

- If you would like to receive our 'Research Roundup', a monthly newsletter highlighting the cutting-edge research published in The Physiological Society's family of journals (The Journal of Physiology, Experimental Physiology, Physiological Reports, The Journal of Nutritional Physiology and The Journal of Precision Medicine: Health and Disease), please click this link, fill in your name and email address and select 'Research Roundup': <https://www.physoc.org/journals-and-media/membernews>

EDITOR COMMENTS

Reviewing Editor:

Thank you for responding to the outstanding comments. There are no further comments for you to consider.